# A reservoir of timescales emerges in recurrent circuits with heterogeneous neural assemblies

**Merav Stern[1,2†], Nicolae Istrate[1,3†], Luca Mazzucato[1,3,4]\***

[1]Institute of Neuroscience, University of Oregon, Eugene, United States; [2]Faculty of Medicine, The Hebrew University of Jerusalem, Jerusalem, Israel; [3]Departments of Physics, University of Oregon, Eugene, United States; [4]Mathematics and Biology, University of Oregon, Eugene, United States

**Abstract** The temporal activity of many physical and biological systems, from complex networks to neural circuits, exhibits fluctuations simultaneously varying over a large range of timescales. Long-tailed distributions of intrinsic timescales have been observed across neurons simultaneously recorded within the same cortical circuit. The mechanisms leading to this striking temporal heterogeneity are yet unknown. Here, we show that neural circuits, endowed with heterogeneous neural assemblies of different sizes, naturally generate multiple timescales of activity spanning several orders of magnitude. We develop an analytical theory using rate networks, supported by simulations of spiking networks with cell-type specific connectivity, to explain how neural timescales depend on assembly size and show that our model can naturally explain the long-tailed timescale distribution observed in the awake primate cortex. When driving recurrent networks of heterogeneous neural assemblies by a time-dependent broadband input, we found that large and small assemblies preferentially entrain slow and fast spectral components of the input, respectively. Our results suggest that heterogeneous assemblies can provide a biologically plausible mechanism for neural circuits to demix complex temporal input signals by transforming temporal into spatial neural codes via frequency-selective neural assemblies.

**\*For correspondence:**
lmazzuca@uoregon.edu

[†]These authors contributed equally to this work

**Competing interest:** The authors declare that no competing interests exist.

## Editor's evaluation

This fundamental work uses computational network models to suggest a possible origin of the wide range of time scales observed in cortical activity. This claim is supported by convincing evidence based on comparisons between mathematical theory, simulations of spiking network models, and analysis of recordings from the orbitofrontal cortex. This manuscript will be of interest to the broad community of systems and computational neuroscience.

## Introduction

Experimental evidence shows that the temporal activity of many physical and biological systems exhibits fluctuations simultaneously varying over a large range of timescales. In condensed matter physics, spin glasses typically exhibit aging and relaxation effects whose timescales span several orders of magnitude (*Bouchaud, 1992*). In biological systems, metabolic networks of *E. coli* generate fluxes with a power-law distribution of rates (*Almaas et al., 2004*; *Emmerling et al., 2002*). Gas release in yeast cultures exhibits frequency distributions spanning many orders of magnitude (*Roussel and Lloyd, 2007*), endowing them with robust and flexible responses to the environment (*Aon et al., 2008*). In the mammalian brain, a hierarchy of timescales in the activity of single neurons is observed

across different cortical areas from occipital to frontal regions (*Murray et al., 2014*; *Siegle et al., 2019*; *Gao et al., 2020*). Moreover, neurons within the same local circuit exhibit a large range of timescales from milliseconds to minutes (*Bernacchia et al., 2011*; *Cavanagh et al., 2016*; *Miri et al., 2011*). This heterogeneity of neuronal timescales was observed in awake animals during periods of ongoing activity in the absence of external stimuli or behavioral tasks, suggesting that long-tailed distributions of intrinsic timescales may be an intrinsic property of recurrent cortical circuits. Recent studies highlighted the benefits of leveraging computations on multiple timescales when performing complex tasks in primates (*Iigaya et al., 2019*) as well as in artificial neural networks (*Perez-Nieves et al., 2021*). However, the neural mechanisms underlying the emergence of multiple timescales are not yet understood.

Here, we present a simple and robust neural mechanism generating heterogeneous timescales of activity in recurrent circuits. The central feature of our model is a heterogeneous distribution of cell assemblies, a common ingredient observed in cortical architecture (*Perin et al., 2011*; *Marshel et al., 2019*; *Figure 1a*). We first demonstrate that rate networks, whose rate units represent cell-assemblies, can generate long-tailed distributions of timescales when endowed with heterogeneous assemblies (*Figure 1b*). We then show that the heterogeneity of timescales, observed in electrophysiological recordings from awake primate cortex (*Cavanagh et al., 2016*), can be explained by the presence of heterogeneous cell assemblies (*Figure 1b*). Using methods from statistical physics, we develop an analytical framework explaining how an assembly's intrinsic timescale depends on size, revealing the emergence of a new chaotic regime where activity is bistable. We show that our theory applies to biologically plausible models of cortical circuits based on spiking networks with cell-type specific clustered architectures.

We then study the stimulus-response properties of networks with heterogeneous assemblies. In networks with homogeneous assemblies, chaotic activity is suppressed at a single resonant frequency (*Rajan et al., 2010*). However, when we drive heterogeneous networks with a time-dependent broadband input featuring a superposition of multiple frequencies, we find that the chaotic activity is suppressed across multiple frequencies which depend on the assembly own size. Large and small assemblies are preferentially entrained by the low and high-frequency components of the input, respectively (*Figure 1c*). This spectral specificity suggests that a reservoir of timescales may be a natural mechanism for cortical circuits to flexibly demix different spectral features of complex time-varying inputs. This mechanism may endow neural circuits with the ability to transform temporal neural codes into spatial neural codes via frequency-selective neural assemblies.

## Results

To develop a theory of heterogeneous timescales, we first focus on random neuronal networks whose rate units are recurrently connected, with couplings that are chosen randomly. In this model, we will be able to leverage analytical methods from statistical field theory (*Sompolinsky et al., 1988*; *Buice and Chow, 2013*; *Helias and Dahmen, 2020*) to link analytical model parameters to circuit dynamics. In our rate network model, each network unit represents a *functional assembly* of cortical neurons with similar response properties. We interpret the unit's *self-coupling* as the *size* of the corresponding neural assembly (if recurrent couplings across the population vary significantly, we also interpret the *self-coupling* as representing the average coupling strength within an assembly). In the case where the self-couplings are zero or weak (order $1/\sqrt{N}$, with $N$ being the size of the network), random networks are known to undergo a phase transition from silence to chaos when the variance of the random couplings exceeds a critical value (*Sompolinsky et al., 1988*). When the self-couplings are strong (order 1) and are all equal, a third phase appears, featuring multiple stable fixed points accompanied by long transient activity (*Stern et al., 2014*). In all these cases, all network units exhibit the same intrinsic timescale, estimated from their autocorrelation function. Here, we demonstrate a novel class of recurrent networks, capable of generating temporally heterogeneous activity whose multiple timescales span several orders of magnitude. We show that when the self-couplings are heterogeneous, a reservoir of multiple timescales emerges, where each unit's intrinsic timescale depends both on its own self-coupling and the network's self-coupling distribution.

## a)  Recurrent circuits with heterogeneous neural assemblies

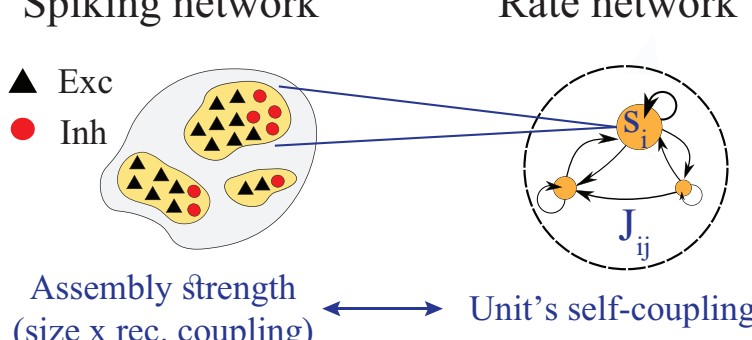

## b)  Long-tailed timescale distribution fits primate cortex

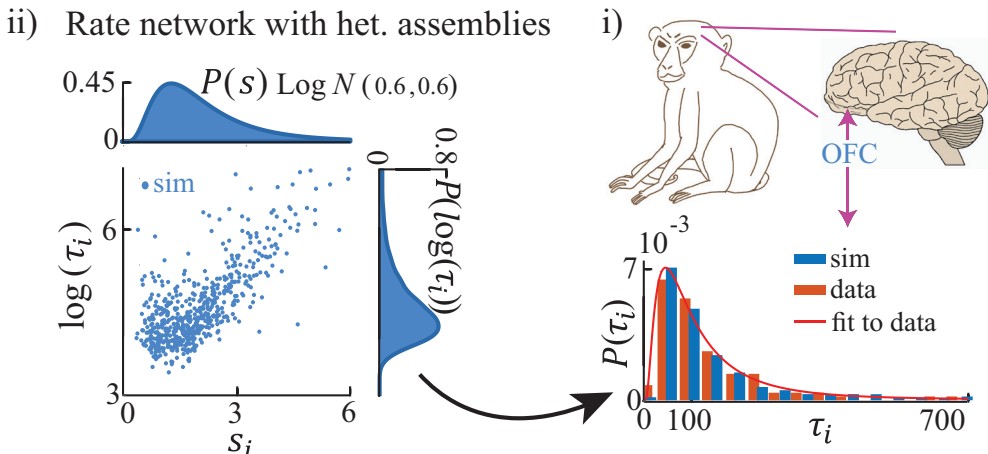

## c)  Demixing input frequencies with heterogeneous assemblies

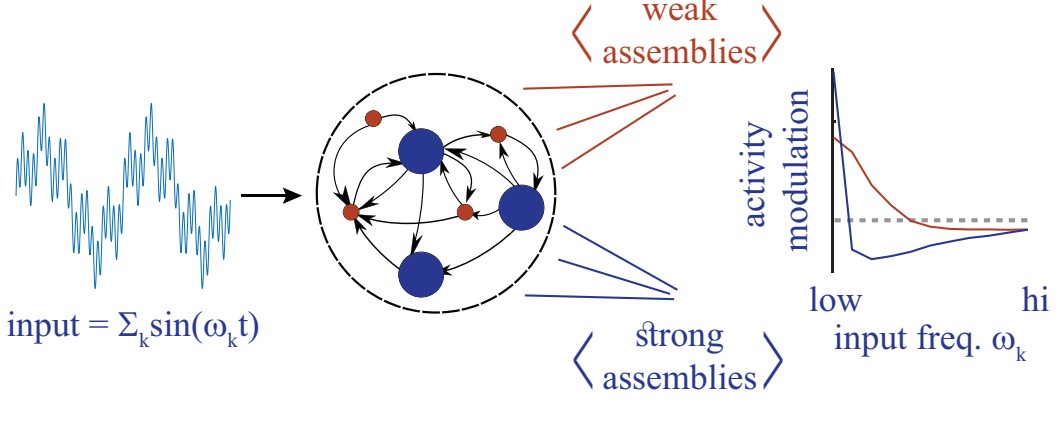

**Figure 1.** Summary of the main results. (**a**) Left: Microscopic model based on a recurrent network of spiking neurons with excitatory and inhibitory cell types, arranged in neural assemblies of heterogeneous sizes. Right: Phenomenological model based on a recurrent network of rate units. Each unit corresponds to an E/I neural assembly, whose size is represented by the unit's self-couplings $s_i$. (**b**) A lognormal distribution of self-couplings (representing assemblies of different sizes) generates time-varying activity whose heterogeneous distribution of timescale fits population activity

*Figure 1 continued on next page*

*Figure 1 continued*

recorded from awake monkey orbitofrontal cortex (data from *Cavanagh et al., 2016*). (**c**) When driving our heterogeneous network with broadband time-varying input, comprising a superposition of sine waves of different frequencies, large and small assemblies preferentially entrain with low and high spectral components of the input, respectively, thus demixing frequencies into responses of different populations.

## Random networks with heterogeneous self-couplings

We start by considering a recurrent network of $N$ rate units obeying the dynamical equations

$$\frac{dx_i}{dt} = -x_i + s_i\phi(x_i) + g\sum_{j=1}^{N} J_{ij}\phi(x_j) \tag{1}$$

where the random couplings $J_{ij}$ from unit $j$ to unit $i$ are drawn independently from a Gaussian distribution with mean 0 and variance $1/N$; $g$ represents the network gain and we chose a transfer function $\phi(x) \equiv \tanh(x)$. The self-couplings $s_i$ are drawn from a distribution $P(s)$. The special case of equal self-couplings ($s_i = s$) was studied by *Stern et al., 2014* and a summary of the results can be found in Appendix 1 for convenience. Here, we study the network properties in relation to both discrete and continuous distributions $P(s)$.

Using standard methods of statistical field theory (*Buice and Chow, 2013*; *Helias and Dahmen, 2020*, see Methods: 'Dynamic mean-field theory with multiple self-coupling' for details), in the limit of large $N$ we can average over realizations of the disordered couplings $J_{ij}$ to derive a set of self-consistent dynamic mean-field equations for each population of units $x_\alpha$ with self-coupling strengths $s_\alpha \in S$

$$\frac{dx_\alpha}{dt} = -x_\alpha + s_\alpha \tanh(x_\alpha) + \eta(t) . \tag{2}$$

In our notation, $S$ denotes the set of different values of self-couplings $s_\alpha$, indexed by $\alpha \in A$, and we denote by $N_\alpha$ the number of units with the same self-coupling $s_\alpha$, and accordingly by $n_\alpha = N_\alpha/N$ their fraction. The mean-field $\eta(t)$ is the same Gaussian process for all units and has zero mean $\langle\eta(t)\rangle = 0$ and autocorrelation

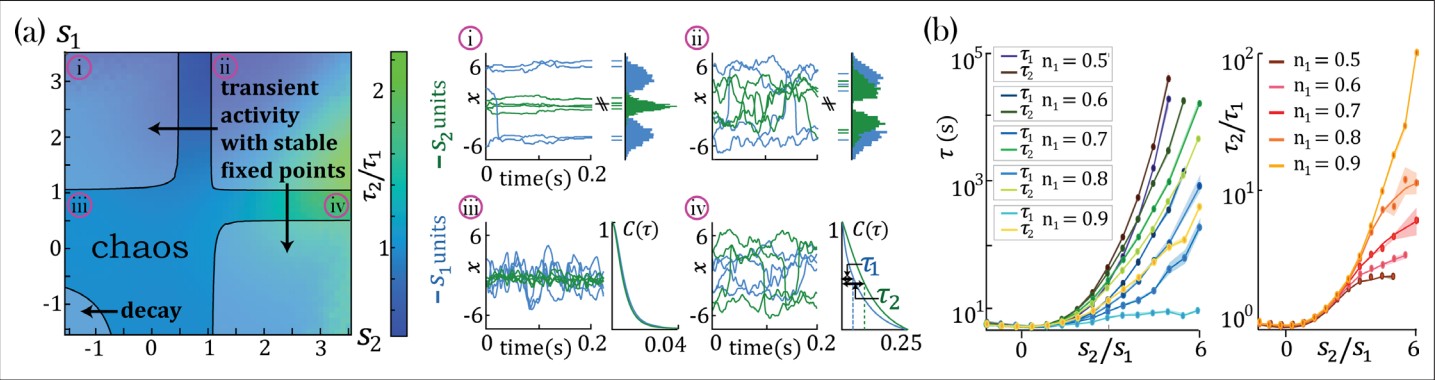

**Figure 2.** Dynamical and fixed point properties of networks with two self-couplings. (**a**) Ratio of autocorrelation timescales $\tau_2/\tau_1$ of units with self-couplings $s_2$ and $s_1$, respectively ($\tau_i$ is estimated as the half width at half max of a unit's autocorrelation function, see panels iii, iv), in a network with $n_1 = n_2 = 0.5$ and $g = 2$ and varying $s_1, s_2$. A central chaotic phase separates four different stable fixed point regions with or without transient activity. Black curves represent the transition from chaotic to stable fixed point regimes, which can be found by solving consistently *Equation 15*, *Equation 16*, and *Equation 18* (using equal to 1 in the latter), see Methods ('Fixed points and transition to chaos' and 'Stability conditions') for details. (**i**, **ii**) Activity across time during the initial transient epoch (left) and distributions of unit values at their stable fixed points (right), for networks with $N = 1000$ and (**i**) $s_1 = 3.2, s_2 = -1.5$, (ii) $s_1 = 3.2, s_2 = 1.2$. (**iii**, **iv**) Activity across time (left) and normalized autocorrelation functions $C(\tau)/C(0)$, (right) of units with (iii) $s_1 = 0.8, s_2 = -1.5$, (iv) $s_1 = 0.8, s_2 = 3.2$. (**b**) Timescales $\tau_2, \tau_1$ (left) and their ratio $\tau_2/\tau_1$ (right) for fixed $s_1 = 1$ and varying $s_2$, as a function of the relative size of the two populations $n_1 = N_1/N, n_2 = N_2/N$ (at $g = 2$, $N = 2000$; average over 20 network realizations). All points in b. were verified to be within the chaotic region using *Equation 18*.

$$\langle \eta(t)\eta(t+\tau)\rangle \quad = g^2 C(\tau)$$
$$C(\tau) \quad = \sum_{\alpha \in A} n_\alpha \langle \phi[x_\alpha(t)]\phi[x_\alpha(t+\tau)]\rangle \,, \tag{3}$$

where $\langle \cdot \rangle$ denotes an average over the mean-field.

We found that networks with heterogeneous self-couplings exhibit a complex landscape of fixed points $x_\alpha^*$, obtained as the self-consistent solutions to the static version of *Equation 2* and *Equation 3*, subject to stability conditions (see Methods: 'Fixed points and transition to chaos' and 'Stability conditions'). For fixed values of the network gain $g$, these fixed points can be destabilized by varying the self-couplings of different assemblies, inducing a transition to time-varying chaotic activity (*Figure 2*). The fixed points landscape exhibits remarkable features inherited directly from the single value *self-coupling* case, as was extensively researched in *Stern et al., 2014*. Here, we focus on the dynamical properties of the time-varying chaotic activity, which constitute new features resulting from the heterogeneity of the *self-couplings*. We illustrate the network's dynamical features in the case of a network with two sub-populations with $n_1$ and $n_2 = 1 - n_1$ portions of the units with self-couplings $s_1$ and $s_2$, respectively. In the $(s_1, s_2)$ plane, this model gives rise to a phase diagram with a single chaotic region separating four disconnected stable fixed-point regions (*Figure 2a*). In the case of a Gaussian distribution of self-couplings in the stable fixed point regime, a complex landscape of stable fixed points emerges. The unit values at the stable fixed points continuously interpolate between around zero (for units with $s_i < 1$) and a bi-modal distribution (for units with $s_i > 1$) within the same network (*Figure 3a*).

## A reservoir of heterogeneous timescales explains cortical recordings

In the chaotic phase, we can estimate the intrinsic timescale $\tau_i$ of a unit $x_i$ from its autocorrelation function $C(\tau) = \langle \phi[x_i(t)]\phi[x_i(t+\tau)]\rangle_t$ as the half-width at its autocorrelation half maximum (*Figure 2a-iv*, $\tau_1$, and $\tau_2$). The chaotic phase in the network, *Equation 1*, is characterized by a large range of timescales that can be simultaneously realized across the units with different self-couplings. In a network with two self-couplings $s_1$ and $s_2$ in the chaotic regime, we found that the ratio of the timescales $\tau_2/\tau_1$ increases as we increase the self-couplings ratio $s_2/s_1$ (*Figure 2b*). The separation of timescales depends on the relative fractions $n_1$ and $n_2 = 1 - n_1$ of the fast and slow populations, respectively. When the fraction of $n_2$ approaches zero, (with $n_1 \to 1$), the log of the timescale ratio exhibits a supralinear dependence on the self-couplings ratio, as described analytically in Methods ('Universal colored-noise approximation to the Fokker-Planck theory'), with a simplified estimation given in *Equation 4*, leading to a vast separation of timescales. Other self-couplings ratios $s_2/s_1$ approach the timescale supralinear separation as the fraction of $n_1$ increases. We note that all uses of 'log' to evaluate the timescale growth and otherwise assume the base e.

In the case of a lognormal distribution of self-couplings, in the chaotic regime the network generates a reservoir of multiple timescales $\tau_i$'s of chaotic activity across network units, spanning across several orders of magnitude (*Figure 3b*). For long-tailed distributions such as the lognormal, mean-field theory can generate predictions for rare units with large self-couplings from the tail end of the distribution by solving *Equation 2* and the continuous version of *Equation 3*, see Methods ('Dynamic mean-field theory with multiple self-couplings') *Equation 13*. The solution highlights the exponential relation between a unit's self-coupling and its autocorrelation decay time (*Figure 3aii*, purple dashed line).

During periods of ongoing activity, the distribution of single-cell autocorrelation timescales in primate cortex was found to be right-skewed and approximately lognormal, ranging from 10 ms to a few seconds (*Cavanagh et al., 2016*; *Figure 3bi*). Can the reservoir of timescales generated in our heterogeneous network model explain the distribution of timescales observed in primate cortex? We found that a model with a lognormal distribution of self-couplings can generate a long-tailed distribution of timescales which fits the distribution observed in awake primate orbitofrontal cortex (*Figure 3bi*). This result shows that neural circuits with heterogeneous assemblies can naturally generate the heterogeneity in intrinsic timescales observed in cortical circuits from awake primates.

## Separation of timescales in the bistable chaotic regime

To gain an analytical understanding of the parametric separation of timescales in networks with heterogeneous self-couplings, we consider the special case of a network with two self-couplings

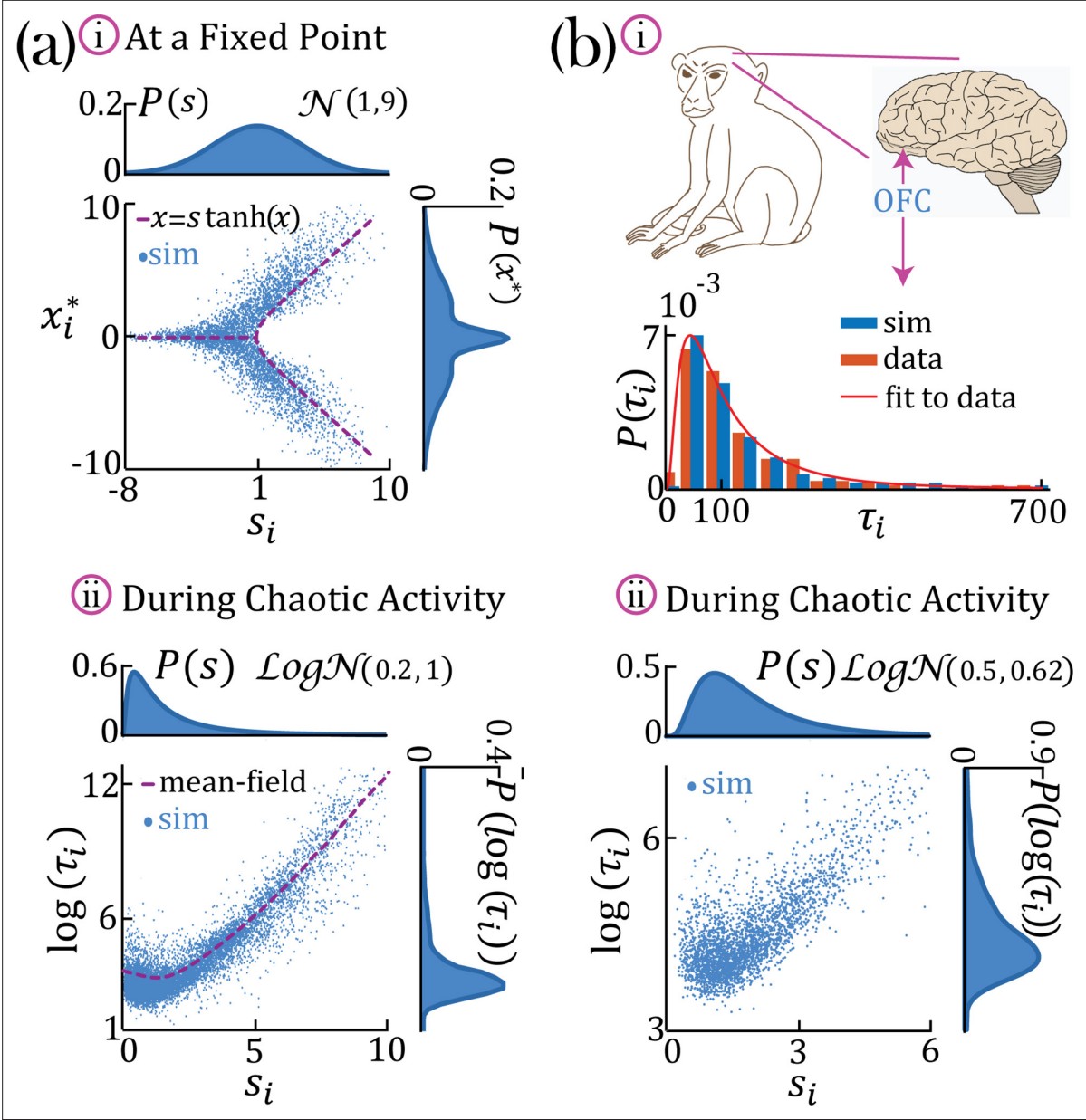

**Figure 3.** Continuous distributions of self-couplings. (**a–i**) In a network with a Gaussian distribution of self-couplings (mean $\mu = 1$ and variance $\sigma^2 = 9$), and $g = 2.5$, the stable fixed point regime exhibits a distribution of fixed point values interpolating between around the zero fixed point (for units with $s_i \leq 1$) and the multi-modal case (for units with $s_i > 1$). The purple curve represents solutions to $x = s \tanh(x)$. (**a,b–ii**) A network with a lognormal distribution of self-couplings (parameters for (a,b) $\mu = 0.2, 0.5$ and $\sigma^2 = 1, 0.62$, and $g = 2.5$ ;autocorrelation timescales $\tau_i$ in units of ms) in the chaotic phase, span several orders of magnitude as functions of the units' self-couplings $s_i$. (a-ii) Mean-field predictions for the autocorrelation functions and their timescales (purple curve) were generated from *Equation 13* and *Equation 14* via an iterative procedure, see Methods: 'Dynamic mean-field theory with multiple self-couplings' , 'An iterative solution'. (**b**) Populations of neurons recorded from orbitofrontal cortex of awake monkeys exhibit a lognormal distribution of intrinsic timescales (data from *Cavanagh et al., 2016*) (panel **b-i**, red), consistent with neural activity generated by a rate network with a lognormal distribution of self-couplings (panel **b-i**, blue; panel **b-ii**). ; We note that *Cavanagh et al., 2016* use fitted exponential decay time constants of the autocorrelation functions as neurons' timescales, while we use the half widths at half max of the autocorrelation functions as the timescales. To bridge these two definitions, we multiplied (*Cavanagh et al., 2016*) data by a factor of ln(2) before comparing it with our model and presenting it in this figure. The model membrane time constant was assumed to be 3 ms in this example.

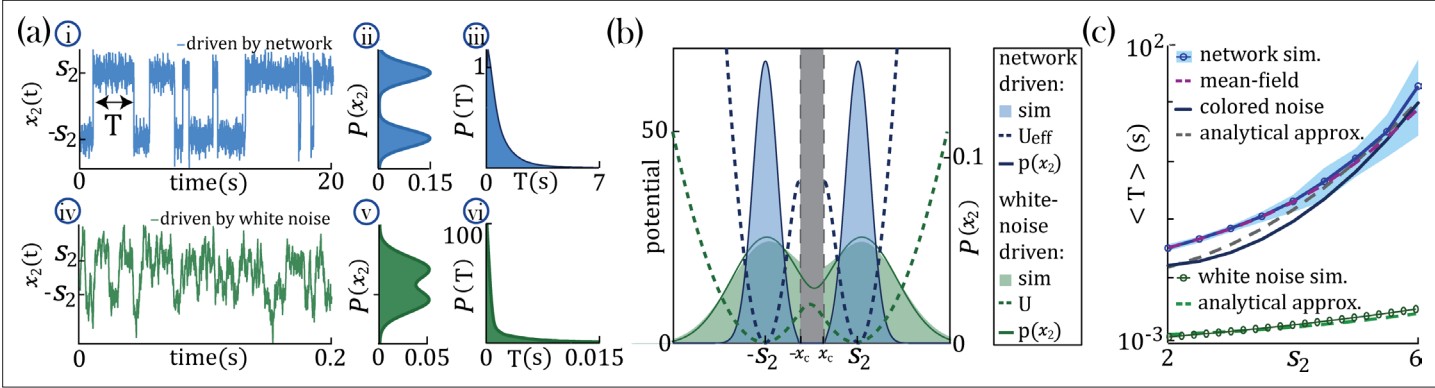

**Figure 4.** Separation of timescales and metastable regime. (**a**) Examples of bistable activity. (**i, iv,i**) - time courses; (**ii, v**) - histograms of unit's value across time; (**iii, vi**) - histograms of dwell times. (**a–i, ii, iii**) An example of a probe unit $x_2$ with $s_2 = 5$, embedded in a neural network with $N = 1000$ units, $N_1 = N - 1$ units with $s_1 = 1$ and $g = 1.5$. (**a–iv, v, vi**) An example of a probe unit driven by white noise. Note the differences in the x-axis scalings of the timecourses (**a–i** vs. **a–iv** and **a–iii** vs. **a–vi**).(**b**) The unified colored noise approximation stationary probability distribution (dark blue curve, **Equation 19**, its support excludes the shaded gray area) from the effective potential $U_{eff}$ (dashed blue curve) captures well the activity histogram (light blue area; same as (**a–ii**)); whereas the white noise distribution (dark green curve, obtained from the naive potential $U$, dashed green curve) captures the probe unit's activity (light green area; same as (**a–v**)) when driven by white noise, and deviates significantly from the activity distribution when the probe is embedded in our network. (**c**) Average dwell times,$\langle T \rangle$, in the bistable states. Simulation results, mean, and 95%CI (blue curve and light blue background, respectively; An example of the full distribution of the dwell times is given in (**a–iii**)). Mean-field predictions (purple curve) were generated by calculating the average dwell times from a trace of $x_2$, which was produced by solving the mean-field equations; **Equation 2** simultaneously and consistently with **Equation 3** with $n_1 = 1$ and $n_2 = 0$. The mean first passage time from the unified colored noise approximation (**Equation 22**, black curve), and for a simplified estimate thereof (**Equation 4**, gray dashed line) capture well $\langle T \rangle$. When driven by white noise (green curve and light green curve are simulation results and simplified estimate using Equation 4, respectively), the probe's average dwell times are orders of magnitude shorter than with colored noise, exhibiting substantial support of the probe distribution in the region where the crossing between wells happens (allowing frequent crossing,(**a–iv**) green line at $x = 0$) and, equivalently, the low value of the potential around its maxima ((**b**) green dashed line at $x = 0$). Comparison of white and colored noise demonstrates the central role of the self-consistent colored noise to achieve long dwell times.

The online version of this article includes the following figure supplement(s) for figure 4:

**Figure supplement 1.** Validation of universal colored noise approximation (UCNA) approach to estimate escape times.

where a large sub-population ($N_1 = N - 1$) with $s_1 = 1$ comprises all but one slow probe unit, $x_2$, with large self-coupling $s_2 \gg s_1$ (see Methods:'Universal colored-noise approximation to the Fokker-Planck theory' for details). In the large $N$ limit, we can neglect the backreaction of the probe unit on the mean-field and approximate the latter as an external Gaussian colored noise $\eta(t)$ with autocorrelation $g^2 C(\tau) = g^2 \langle \phi[x_1(t)]\phi[x_1(t + \tau)]\rangle$, independent of $x_2$. The noise $\eta(t)$ then represents the effect on the probe unit $x_2$ of all other units in the network and can be parameterized by the noise strength $D$ and its timescale (color) $\tau_1$. For large $s_2$, the dynamics of the probe unit $x_2$ leads to the emergence of a bi-stable chaotic phase whereby its activity is localized around the critical points $x^{\pm} \simeq \pm s_2$ (**Figure 4a–i**) and switches between them at random times. In the regime of colored noise (as we have here, with $\tau_1 \simeq 7.9 \gg 1$), the stationary probability distribution $p(x_2)$ (**Figure 4a–ii and b**) satisfies the unified colored noise approximation to the Fokker Planck equation (see Methods:'Universal colored-noise approximation to the Fokker-Planck theory', **Hänggi and Jung, 1995**; **Jung and Hänggi, 1987**), based on an analytical expression for its effective potential $U_{eff}(x)$ as a function of the self-coupling $s_2$ and the noise color ($\tau_1$). The distribution $p(x_2)$ is concentrated around the two minima $x^{\pm} \simeq \pm s_2$ of $U_{eff}$. The main effect of the strong color $\tau_1 \gg 1$ is to sharply decrease the variance of the distribution around the minima $x^{\pm}$, compared to the white noise case ($\tau_1 = 0$). This is evident from comparing the colored noise with white noise (**Figure 4a-iv,v,vi**).

In our network with colored noise, the probe unit's temporal dynamics are captured by the mean first passage time $\langle T \rangle$ for the escape out of the potential well defined by the effective potential $U_{eff}$, yielding good agreement with simulations at increasing $s_2$, as expected on theoretical ground (**Hänggi and Jung, 1995**; **Jung and Hänggi, 1987**; **Figure 4c**). The asymptotic scaling of the mean first passage time for large $s_2$ is

$$\log(\langle T \rangle) \sim \frac{\tau_1 + 1}{2D}\left[s_2^2 - s_2 \log(s_2)\right] . \tag{4}$$

In this slow probe regime, we thus achieved a parametric separation of timescales between the population $x_1$, with its intrinsic timescale $\tau_1$, and the probe unit $x_2$ whose activity fluctuations exhibit two separate timescales: the slow timescale $<T>$ of the dwelling in each of the bistable states and the fast timescale $\tau_1$ of the fluctuations around the metastable states (obtained by expanding the dynamical equation around the meta-stable values $x^{\pm} = \pm s_2$). One can generalize this metastable regime to a network with $N - p$ units which belong to a group with $s_1 = 1$ and $p \ll N$ slow probe units $x_\alpha$, for $\alpha = 2, \ldots, p + 1$, with large self-couplings $s_\alpha$. The slow dynamics of each probe unit $x_\alpha$ is captured by its own mean first passage time (between the bistable states) $<T>_\alpha$ in (*Equation 22*) and all slow units are driven by a shared external colored noise $\eta(t)$ with timescale $\tau_1$. In summary, in our model multiple timescales can be robustly generated with specific values, varying over several orders of magnitude.

Is the relationship between the unit's self-coupling and its timescale relying on single-unit properties, or does it rely on network effects? To answer this question, we compare the dynamics of a unit when driven by a white noise input vs. the self-consistent input generated by the rest of the recurrent network (i.e. the mean-field). If the neural mechanism underlying the timescale separation was a property of the single-cell itself, we would observe the same effect regardless of the details of the input noise. We found that the increase in the unit's timescale as a function of $s_2$ is absent when driving the unit with white noise, and it only emerges when the unit is driven by the self-consistent mean-field (*Figure 4c*). We thus concluded that this neural mechanism is not an intrinsic property of a single unit but requires the unit to be part of a recurrently connected network.

## A reservoir of timescales in E-I spiking networks

We next investigated whether the neural mechanism endowing the rate network (1) with a reservoir of timescales could be implemented in a biologically plausible model based on spiking activity and excitatory/inhibitory cell-type specific connectivity. To this end, we modeled the local cortical circuit as a recurrent network of excitatory (E) and inhibitory (I) current-based leaky integrated-and-fire neurons (see Methods:'Spiking network model' for details), where both E and I populations were arranged in neural assemblies (*Figure 5a*; *Amit and Brunel, 1997*; *Litwin-Kumar and Doiron, 2012*; *Wyrick and Mazzucato, 2021*). Synaptic couplings between neurons in the same assembly were potentiated compared to those between neurons in different assemblies. Using mean-field theory, we found that the recurrent interactions of cell-type specific neurons belonging to the same assembly can be interpreted as a self-coupling, expressed in terms of the underlying network parameters as $s_i^E = \bar{J}_{EE}^{(in)} C_i^E$, where $C_i^E$ is the assembly size and $\bar{J}_{EE}^{(in)}$ is the average synaptic coupling between E neurons within the assembly (see Methods: 'Spiking network model' for details). The spiking network time-varying activity unfolds through sequences of metastable attractors (*Litwin-Kumar and Doiron, 2012*; *Wyrick and Mazzucato, 2021*), characterized by the co-activation of different subsets of neural assemblies (*Figure 5a*). These dynamics rely on the bistable activity of each assembly, switching between high and low firing rate states. The dwell time of an assembly in a high-activity state increases with larger sizes and with stronger intra-assembly coupling strength (*Figure 5b*). This metastable regime in spiking networks is similar to the bi-stable, heterogeneous timescales activity observed in the random neural networks endowed with heterogeneous self-couplings. We further examined the features of the metastable regime in spiking networks in order to compare the mechanism underlying the heterogeneous timescale distributions in the rate and spiking models. The two models exhibit clear differences in their "building blocks". In the rate network, the transfer function is odd ($\tanh$) leading to bistable states with values localized around $x^{\pm} \simeq \pm s$. In the spiking model, the single neuron current-to-rate transfer function is strictly positive so that the bistable states can have both high and low firing rate values. As a result, in the spiking network, unlike in the rate network, a variety of firing rate levels can be attained during the high activity epochs, depending on both the assembly size and the the number of other simultaneously active assemblies. In other words, the rate level depends on the specific network attractor visited within the realization of the complex attractor landscape (*Mazzucato et al., 2015*; *Wyrick and Mazzucato, 2021*).

Despite these differences, we found crucial similarities between the time-varying activity in the two models related to the underlying neural mechanism. The characteristic timescale $T$ of the assembly metastable dynamics can be estimated from its average activation time (*Figure 5a and b*). We tested whether the heterogeneity in the assembly self-coupling distribution could lead to a

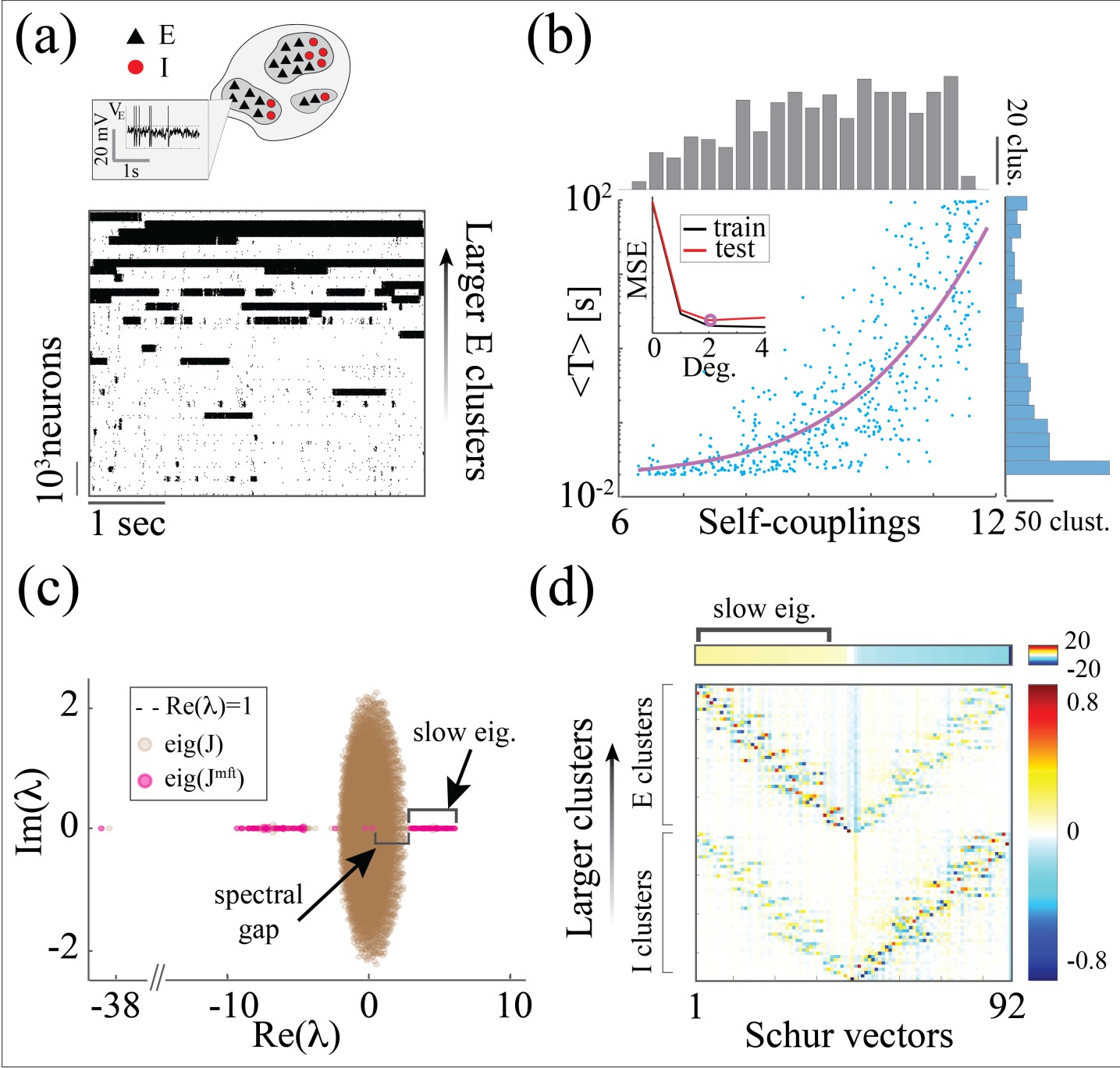

**Figure 5.** *Heterogeneity of timescales in E-I spiking networks.* (**a**) Top: Schematic of a spiking network with excitatory (black) and inhibitory populations (red) arranged in assemblies with heterogeneous distribution of sizes. Bottom: In a representative trial, neural assemblies activate and deactivate at random times generating metastable activity (one representative E neuron per assembly; larger assemblies on top; representative network of $N = 10{,}000$ neurons), where larger assemblies tend to activate for longer intervals. (**b**) The average activation times <T> of individual assemblies (blue dots; the average was calculated across 100s simulation and across all neurons within the same assembly for all assemblies in 20 different network realizations; self-coupling units are in [mV], see Methods section). Fit of $\log(T) = a_2 s_E^2 + a_1 s_E + a_0$ with $a_2 = 0.14, a_1 = 1.97, a_0 = 5.51$ (pink curve). Inset: cross-validated model selection for polynomial fit. As the assembly strength (i.e. the product of its size and average recurrent coupling) increases, <T> increases, leading to a large distribution of timescales ranging from 20 ms to 100s. (**c**) Eigenvalue distribution of the full weight matrix $J$ (brown) and the mean-field-reduced weight matrix $J^{MF}$ (pink). (**d**) The Schur eigenvectors of the weight matrix $J^{MF}$ show that the slow (gapped) Schur eigenvalues (top) are associated with eigenvectors corresponding to E/I cluster pairs (bottom). See Appendix (**e**) Spiking network model for more details and for the scaling to larger networks.

heterogeneous distribution of timescales. We endowed the network with a heterogeneous distribution of assembly sizes (an additional source of heterogeneity originates from the Erdos-Renyi connectivity), yielding a heterogeneous distribution of self-couplings (*Figure 5b*). We found that the assembly activation timescales $T$ grew as the assembly's self-coupling increased, spanning overall a large range of timescales from 20 ms to 100 s i.e. the whole range of our simulation epochs (*Figure 5b*). In particular, the functional dependence of $\log(T)$ vs. self-coupling $s_E$ was best fit by a quadratic polynomial (*Figure 5b* inset, see Methods: 'Spiking network model' for details), in agreement with the functional dependence obtained from the analytical calculation in the rate model (4). We thus concluded that a reservoir of timescales can naturally emerge in biologically plausible spiking models of cortical circuits from a heterogeneous distribution of assembly sizes. Both the range of timescales (20 ms-100 s) (*Cavanagh et al., 2016*) and the distribution of assembly sizes (50–100 neurons) (*Perin et al., 2011*; *Marshel et al., 2019*) are consistent with experimental observations.

What is the relationship between the distribution of timescales in the E/I spiking model and the chaotic rate network? We can obtain crucial insights by combining analysis of the synaptic weight matrix together with a mean-field approach in a linear approximation, following the approach of *Murphy and Miller, 2009*; *Schaub et al., 2015*. In the spiking network, the non-normal weight matrix $J$ exhibits the typical E/I structure with the four submatrices representing E/I cell-type specific connectivity; within each of the four E/I submatrices, diagonal blocks highlight the paired E/I clustered architecture (the heterogeneous distribution of cluster sizes is manifest in the increasing size of the diagonal blocks, *Appendix 2—figure 1a*). To interpret the dynamical features emerging from this weight matrix, we examined a mean-field reduction of the $N$-dimensional network to a set of $2p + 2$ mean-field variables, representing the $2p$ E and I clusters plus the two unclustered background E and I populations (see Appendix 2). The $2p + 2$ eigenvalues of the mean-field-reduced weight matrix $J^{MF}$ comprise a subset of the full weight matrix $J$, capturing the salient features of the spiking network dynamics in a linear approximation (*Figure 5c*; see Appendix 2 and *Murphy and Miller, 2009*; *Schaub et al., 2015*). The weights matrix $J^{MF}$ exhibits a spectral gap, beyond which a distribution of $p - 1$ eigenvalues with real parts larger than one correspond to slow dynamical modes. To identify these slow modes, we examined the Schur eigenvectors of $J^{MF}$, which represent independent dynamical modes in the linearized theory (i.e. an orthonormal basis; see Appendix 2 and *Appendix 2—figure 2*).

We found that the Schur eigenvectors associated with those large positive eigenvalues can be approximately mapped onto E/I cluster pairs. More specifically, eigenvalues with increasingly larger values correspond to assemblies of increasingly larger sizes (*Figure 5c*), which, in turn, are associated with slower timescales (*Figure 5b*). We conclude that the slow switching dynamics in the spiking network is linked to large positive eigenvalues of the synaptic weight matrix, and the different timescales emerge from a heterogeneous distribution of these eigenvalues. For comparison, in the chaotic rate network, the eigenvalue distribution of the weight matrix exhibits a set of eigenvalues with large positive real parts as well, *Appendix 2—figure 3a*. The relation between the value of an eigenvalue and the slow dynamics holds in the rate networks as well: increasingly larger eigenvalues correspond to increasingly larger cluster self-couplings (*Appendix 2—figure 3*) which are associated with slower dynamics (*Figure 3aii*). Therefore, the dynamical structure in the rate networks qualitatively matches the structure uncovered in the mean-field-reduced spiking network weight matrix $J^{MF}$ (*Figure 5C*). In summary, our analysis shows that in both spiking and rate networks the reservoir of timescales is associated with the emergence of a heterogeneous distribution of large positive eigenvalues in the weight matrix. This analysis suggests that the correspondence between rate networks and spiking networks should be performed at the level of dynamical modes associated with these large positive eigenvalues in the Schur basis, where rate units in the rate network can be related to E/I cluster pairs (Schur eigenvectors) in the spiking network. A potential difference between the two models may be related to the nature of the transitions between bistable states in the rate network vs the transitions between low and high activity states in the spiking network. In the rate network, transitions are driven by the self-consistent colored noise, the hallmark of the chaotic activity arising from the disorder in the synaptic couplings. In the spiking network, although the disorder in the inter-assembly effective couplings may contribute to the state transitions, there might be finite size effects at play, due to the small assembly size.

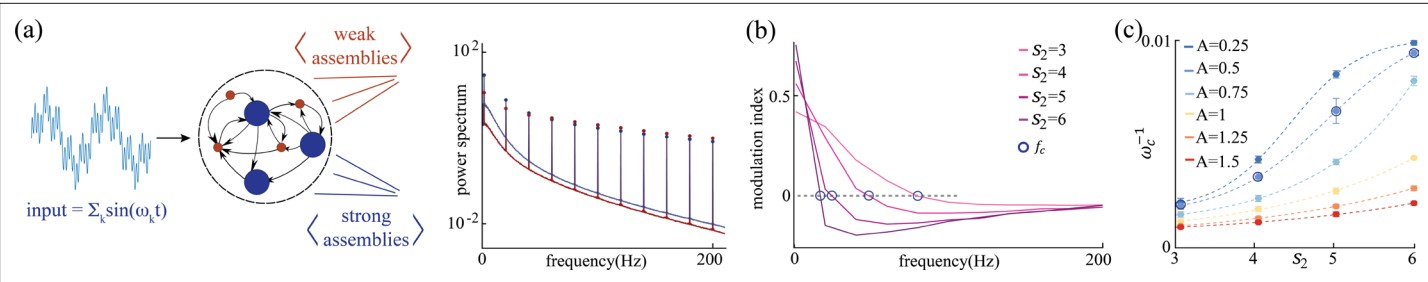

**Figure 6.** Network response to broadband input. (**a**) Power spectrum density of a network driven by time-dependent input comprising a superposition of 11 sinusoidal frequencies (see main text for details). Maroon and navy curves represent average power spectrum density in $s_1$ and $s_2$ populations, respectively; circles indicate the peak in the power spectrum density amplitudes at each frequency; amplitude A = 0.5, g = 3.0, $s_1 = 1$, and $s_2 = 4$. (**b**) Modulation index of the power spectrum density amplitudes as a function of frequency in networks with $s_1 = 1$ and various $s_2$. The blue circles mark the cutoff frequency $f_c$ where the modulation index changes sign. (**c**) Cutoff period, $2\pi\omega_c^{-1}$, as a function of self-coupling $s_2$ for different input amplitudes. An inversely proportional relation between the cutoff period and the amplitude of the broadband signal is present.

## Spatial de-mixing of time-varying broadband input

What are the computational benefits of having multiple timescales simultaneously operating in the same circuit? Previous work in random networks with no self-couplings ($s_i = 0$ in Equation 1) showed that stimulus-driven suppression of chaos is enhanced at a particular input frequency, related to the network's intrinsic timescale (**Rajan et al., 2010**). The phenomenon was preserved when a single rate of adaptation was added to all units (**Muscinelli et al., 2019**). We investigated whether, in our network with two different self-couplings $s_1 < s_2$ (in the chaotic regime), the stimulus-dependent suppression of chaos exhibited different features across the two sub-populations, depending on their different intrinsic timescale. We drove each network unit $x_i$ with an external broadband stimulus $I_i(t) = A \sum_{l=1}^{L} \sin(2\pi f_l t + \theta_i)$ consisting of the superposition of $L$ sinusoidal inputs of different frequencies $f_l$ in the range $1 - 200$ Hz, with an equal amplitude $A = 0.5$ and random phases $\theta_i$. We found that the sub-population with a slow, or fast, intrinsic timescale preferentially entrained its activity with slower, or faster, spectral components of the broadband stimulus, respectively (**Figure 6a**). We quantified this effect using a spectral modulation index $m(f) = [(P_2(f) - P_1(f))/(P_2(f) + P_1(f))]$, where $P_\alpha(f)$ is the power-spectrum peak of sub-population $\alpha$ at the frequency $f$ (**Figure 6b**). A positive, or negative, value of $m(f)$ reveals a stronger, or weaker, respectively, entrainment at frequency $f$ in the sub-population $s_2$ compared to $s_1$ exhibited a crossover behavior whereby the low frequency component of the input predominantly entrained the slow population $s_2$, while the fast component of the input predominantly entrained the fast population $s_1$. When fixing $s_1 = 1$ and varying $s_2$, we found that the dependence of the crossover frequency $f_c$ on $s_2$ was strong at low input amplitudes and was progressively tamed at larger input amplitudes (**Figure 6c**). This is consistent with the fact that the input amplitude completely suppresses chaos beyond a certain critical value, as previously reported in network's with no self-couplings (**Rajan et al., 2010**) and with adaptation (**Muscinelli et al., 2019**).

## Discussion

We demonstrated a new robust and biologically plausible network mechanism whereby multiple timescales emerge across units with heterogeneous self-couplings. In our model, units are interpreted as neural assemblies consistent with experimental evidence from cortical circuits (**Perin et al., 2011**; **Lee et al., 2016**; **Kiani et al., 2015**; **Miller et al., 2014**; **Marshel et al., 2019**), and previous theoretical modeling (**Litwin-Kumar and Doiron, 2012**; **Wyrick and Mazzucato, 2021**). We found that the neural mechanism underlying the large range of timescales is the heterogeneity in the distribution of self-couplings (representing neural assembly size). We showed that this mechanism can be naturally implemented in a biologically plausible model of a neural circuit based on spiking neurons with excitatory/inhibitory cell-type specific connectivity. This spiking network represents a microscopic realization of our mechanism where neurons are arranged in assemblies, and an assembly's self-coupling represents the strength of the recurrent interactions between neurons belonging to that assembly, proportional to its size. A heterogeneous distribution of assembly sizes, in turn, generates a reservoir

of timescales. Crucially, our model captured the distribution of intrinsic timescales observed across neurons recorded within the same area in primate cortex (*Cavanagh et al., 2016*).

Several experimental studies uncovered heterogeneity of timescales of neural activity across brain areas and species. Comparison of the population-averaged autocorrelations across cortical areas revealed a hierarchical structure, varying from 50 ms to 350 ms along the occipital-to-frontal axis (*Murray et al., 2014*). Neurons within the same area exhibit a wide distribution of timescales as well. A heterogeneous distribution of timescales (from 0.5 s to 50 s) was found across neurons in the oculomotor system of the fish (*Miri et al., 2011*) and primate brainstem (*Joshua et al., 2013*), suggesting that timescale heterogeneity is conserved across phylogeny. During periods of ongoing activity, the distribution of single-cell autocorrelation timescales in primates was found to be right-skewed and approximately lognormal, ranging from 10 ms to a few seconds (*Cavanagh et al., 2016* and *Figure 3*). Single neuron activity was found to encode long reward memory traces in primate frontal areas over a wide range of timescales up to 10 consecutive trials (*Bernacchia et al., 2011*). In these studies, autocorrelation timescales were estimated using parametric fits, which may be affected by statistical biases, although Bayesian generative approaches might overcome this issue (*Zeraati et al., 2020*). In our model, we estimated timescales nonparametrically as the half-width at half-maximum of the autocorrelation function. In our biologically plausible model based on a spiking network with cell-type specific connectivity, the distribution of timescales was in the range between 20 ms and 100 s, similar to the range of timescales observed in experiments (*Miri et al., 2011*; *Joshua et al., 2013*; *Cavanagh et al., 2016*). Moreover, the distribution of assembly sizes in our model is within the range of 50–100 neurons, consistent with the size of functional assemblies experimentally observed in cortical circuits (*Perin et al., 2011*; *Marshel et al., 2019*). A fundamental new prediction of our model, to be tested in future experiments, is the direct relationship between assembly size and its timescale.

Previous neural mechanisms for generating multiple timescales of neural activity relied on single-cell bio-physical properties, such as membrane or synaptic time constants (*Gjorgjieva et al., 2016*). In feedforward networks, developmental changes in single-cell conductance can modulate the timescale of information transmission, explaining the transition from slow waves to rapid fluctuations observed in the developing cortex (*Gjorgjieva et al., 2014*). However, the extent to which this single-cell mechanism might persist in the presence of strong recurrent dynamics was not assessed. To elucidate this issue, we examined whether a heterogeneous distribution of single-unit integration time constants could lead to a separation of timescales in a random neural network (see Appendix 3 for details). In this model, half of the units were endowed with a fixed fast time constant and the other half with a slow time constant, whose value varied across networks. We found that, although the average network timescale increased proportionally to the value of the slower time constants, the difference in autocorrelation time between the two populations remained negligible. These results suggest that, although the heterogeneity in single-cell time constants may affect the dynamics of single neurons in isolation or within feedforward circuits (*Gjorgjieva et al., 2014*), the presence of strong recurrent dynamics fundamentally alters these single-cell properties in a counterintuitive way. Our results suggest that a heterogeneity in single-cell time constants may not lead to a diversity of timescales in the presence of recurrent dynamics.

Our results further clarified that the relationship between an assembly's self-coupling and its timescale relies on the strong recurrent dynamics. This relationship is absent when driving an isolated assembly with white noise external input (*Figure 4*). Indeed, the mechanism linking the self-coupling to the timescale only emerged when driving the unit with a mean-field whose color was self-consistently obtained from an underlying recurrent network of self-coupled units.

Previous models showed that a heterogeneity of timescales may emerge from circuit dynamics through a combination of structural heterogeneities and heterogeneous long-range connections arranged along a spatial feedforward gradient (*Chaudhuri et al., 2014*; *Chaudhuri et al., 2015*). These networks can reproduce the population-averaged hierarchy of timescales observed across the cortex in the range of 50–350 ms (*Murray et al., 2014*; *Chaudhuri et al., 2015*). A similar network architecture can also reproduce the heterogeneous relaxation time after a saccade, found in the brainstem oculomotor circuit (*Miri et al., 2011*; *Joshua et al., 2013*), in a range between 10–50 s (*Inagaki et al., 2019*; *Recanatesi et al., 2022*). This class of models can explain a timescale separation within a factor of 10, but it is not known whether they can be extended to several orders of magnitude, as observed between neurons in the same cortical area (*Cavanagh et al., 2016*). Moreover, while the

feedforward spatial structure underlying these two models is a known feature of the cortical hierarchy and of the brainstem circuit, respectively, it is not known whether such a feedforward structure is present within a local cortical circuit. Our model, on the other hand, relies on strong recurrent connectivity and local functional assemblies, two hallmarks of the architecture of local cortical circuits (*Perin et al., 2011*; *Lee et al., 2016*; *Kiani et al., 2015*; *Miller et al., 2014*; *Marshel et al., 2019*). Other network models generating multiple timescales of activity fluctuations were proposed based on self-tuned criticality with anti-hebbian plasticity (*Magnasco et al., 2009*), or multiple block-structured connectivity (*Aljadeff et al., 2015*).

In our model, the dynamics of units with large self-couplings, exhibiting slow switching between bistable states, can be captured analytically using the universal colored noise approximation (UCNA) to the Fokker-Planck equation (*Hänggi and Jung, 1995*; *Jung and Hänggi, 1987*). This is a classic tool from the theory of stochastic processes, which we successfully applied to investigate neural network dynamics for the first time. This slow-switching regime may underlie the emergence of metastable activity, ubiquitously observed in the population spiking activity of behaving mammals (*Abeles et al., 1995*; *Jones et al., 2007*; *Mazzucato et al., 2015*; *Mazzucato et al., 2019*; *Recanatesi et al., 2022*; *Engel et al., 2016*; *Kadmon Harpaz et al., 2019*). In these spiking networks, it is not known how to estimate the timescales of metastable activity from network parameters, and we anticipate that our UCNA may provide a powerful new tool for investigating network dynamics in these biologically plausible models.

What is the functional relevance of neural circuits exhibiting a reservoir of multiple timescales? The presence of long timescales deeply in the chaotic regime is a new feature of our model which may be beneficial for memory capacity away from the edge of chaos (*Toyoizumi and Abbott, 2011*). Moreover, we found that, in our model, time-dependent broadband inputs suppress chaos in a population-specific way, whereby populations of large (small) assemblies preferentially entrain slow (fast) spectral components of the input. Previously studied spiking models suggested that preferential entrainment of input is possible by cellular mechanisms (*Lindner, 2016*) or finite-size fluctuations in a feedforward network structure (*Deger et al., 2014*). Here, we presented a recurrent network mechanism for population-specific chaos suppression, independent of the network size. This mechanism may thus grant recurrent networks with a natural and robust tool to spatially demix complex temporal inputs (*Perez-Nieves et al., 2021*) as observed in visual cortex (*Mazzoni et al., 2008*). Third, the presence of multiple timescales may be beneficial for performing flexible computations involving simultaneously fast and slow timescales, such as in role-switching tasks (*Iigaya et al., 2019*) or as observed in time cells in the hippocampus (*Kraus et al., 2013*; *Howard et al., 2014*). A promising direction for future investigation is the exploration of the computational properties of our model in the context of reservoir computing (*Sussillo and Abbott, 2009*) or recurrent networks trained to perform complex cognitive tasks (*Yang et al., 2019*).

## Methods
### Dynamic mean-field theory with multiple self-couplings
We derive the dynamic mean-field theory in the limit $N \to \infty$ by using the moment generating functional (*Sompolinsky and Zippelius, 1982*; *Crisanti and Sompolinsky, 1987*). For the derivation we follow the Martin-Siggia-Rose-De Dominicis-Janssen path integral approach formalism (*Martin et al., 1973*) as appears extensively in *Helias and Dahmen, 2020*; we borrow their notations as well.

The model includes two sets of random variables, the connectivity couplings $J_{ij}$ for $1 \le i, j \le N; i \ne j$, are drawn independently from a Gaussian distribution with variance $\frac{1}{N}$ and mean 0; and the self-couplings $s_i$ for $1 \le i \le N$, whose values are of order 1. When we examine the dynamics governing each unit in *Equation 1*, the sum over the random couplings $J_{ij}$ contributes $N$ terms which, in the limit $N \to \infty$, ensure that the net contribution (mean and variance) from this sum remains of order 1. Hence, in our model, as in previous models, $J$ is the quenched disorder parameter, whose sum gives rise to the mean-field. The self-couplings (one for each unit) contribute an additional term to the moment generating functional. Each unit's activity strongly depends on the value of its own self-coupling, and hence can't be averaged over when we study a unit's dynamics. After averaging over $J$, we can study all units with the same self-coupling together, as they obey the same mean-filed equation, *Equation 2*. Moreover we show that all units, regardless of their self-coupling, obey a single mean-field due to the

structure of $J$. We note that the results of this Methods section, including *Equation 5* and *Equation 6*, will not be affected by diagonal elements in J which are not zero but rather drawn from the same distribution as the off-diagonal elements (as in the main text) since the contribution of such non-zero elements is negligible overall. To maintain the clarity of the text, and since the results are not affected by it, we left out the differentiation between including and excluding diagonal elements of J of order 1/sqrt(N) in the main text.

For our model, *Equation 1*, the moment generating functional is, therefore, given by:

$$Z = \int \mathcal{D}\tilde{x}\mathcal{D}x \exp\left[ \quad \int dt \sum_{i=1}^{N} \tilde{x}_i(t)\left[(\partial_t + 1)x_i(t) - s_i\phi(x_i(t))\right] \right.$$
$$\left. + \sum_{i=1}^{N} \lambda_i(t)x_i(t) - \sum_{j\neq i} \tilde{x}_i(t)J_{ij}\phi(x_j(t))\right], \tag{5}$$

where $\mathcal{D}x = \prod_i \mathcal{D}x_i$ and $\mathcal{D}\tilde{x} = \prod_i \mathcal{D}\tilde{x}_i/2\pi i$. To start, we calculate $\langle Z(J)\rangle_J$. We take advantage of the self-averaging nature of our model, particularly by averaging over the quenched disorder, $J$. The couplings, $J_{ij}$, are i.i.d. variables extracted from a normal distribution and appear only in the last term in (*Equation 5*). We, hence, focus our current calculation step on that term, and we derive the result to the leading term in $N$, yielding:

$$\int \prod_{i\neq j} dJ_{ij} \sqrt{\frac{N}{2\pi g^2}} \exp\left[-\frac{J_{ij}^2 N}{2g^2}\right] \exp\left[-\int dt\,\tilde{x}_i(t)J_{ij}\phi(x_j(t))\right]$$
$$= \exp\left[\frac{1}{2}\int dtdt'\left(\sum_i \tilde{x}_i(t)\tilde{x}_i(t')\right)\left(\frac{g^2}{N}\sum_j \phi(x_j(t))\phi(x_j(t'))\right)\right]. \tag{6}$$

The result above suggests that all the units in our network are coupled to one another equivalently (by being coupled only to sums that depend on all units' activity). To further decouple the network, we define the quantity

$$Q_1(t, t') \equiv \frac{g^2}{N} \sum_j \phi(x_j(t))\phi(x_j(t')).$$

We enforce this definition by multiplying the disordered averaged moment generating functional with the appropriate Dirac delta function, $\delta$, in its integral form:

$$1 = \quad \int \frac{g^2}{N} dQ_1 \delta\left[-\frac{N}{g^2}Q_1 + \sum_j \phi(x_j(t))\phi(x_j(t'))\right]$$
$$= \quad \int \frac{g^2}{N} dQ_1 dQ_2 \exp Q_2\left[-\frac{N}{g^2}Q_1 + \sum_j \phi(x_j(t))\phi(x_j(t'))\right],$$

where $dQ_2$ is an integral over the imaginary axis (including its $1/(2\pi i)$ factor). We can now rewrite the disordered averaged moment generating functional, using (*Equation 6*) to replace its last term, the definition of $Q_1$, and with multiplying the functional by the $\delta$ function above. All together we get:

$$\langle Z(J)\rangle_J \quad = \int \frac{g^2}{N} dQ_1 dQ_2 \exp\left[-\frac{N}{g^2}\int dtdt' Q_1 Q_2 + N\sum_{\alpha\in A} n_\alpha \ln[Z_\alpha]\right],$$
$$Z_\alpha \quad = \int \mathcal{D}\tilde{x}_\alpha \mathcal{D}x_\alpha \exp\left[\int dt\tilde{x}_\alpha(t)\left((\partial_t + 1)x_\alpha(t) - s_\alpha\phi(x_\alpha(t))\right) \right.$$
$$\left. + \frac{1}{2}\int dtdt' \tilde{x}_\alpha(t)Q_1(t, t')\tilde{x}_\alpha(t') + \int dtdt' \phi(x_\alpha(t))Q_2(t, t')\phi(x_\alpha(t'))\right], \tag{7}$$

with $n_\alpha = N_\alpha/N$ the fraction of units with self-couplings $s_\alpha$ across the population, for $\alpha \in A$. In the expression above we made use of the fact that $Q_1$ and $Q_2$, now in a role of auxiliary fields, couple to sums of the fields $x_i^2$ and $\phi_i^2$ and hence the generating functional for $x_i$ and $\tilde{x}_i$ can be factorized with identical multiplications of $Z_\alpha$. Note that in our network, due to the dependency on $s_i$, $x_i$-s are equivalent as long as $s_i$-s are equivalent. Hence, the factorization is for $Z_\alpha$ for all $x_i$ with $s_i = s_\alpha$. Now each factor $Z_\alpha$ includes the functional integrals $\mathcal{D}x_\alpha$ and $\mathcal{D}\tilde{x}_\alpha$ for a single unit with self-coupling $s_\alpha$.

In the large $N$ limit we evaluate the auxiliary fields in (*Equation 7*) by the saddle point approach.e note variable valued at the saddle point by ($*$), obtaining:

$$0 = \frac{\delta}{\delta Q_{1,2}} \left[ -\frac{1}{g^2} \int dt dt' Q_1 Q_2 + \sum_{\alpha \in A} n_\alpha \ln[Z_\alpha] \right] ,$$

and yielding the saddle point values $(Q_1^*, Q_2^*)$:

$$
\begin{aligned}
0 \quad &= -\frac{1}{g^2} Q_1^*(t,t') + \sum_{\alpha \in A} \frac{n_\alpha}{Z_\alpha} \frac{\partial Z_\alpha}{\partial Q_2(t,t')}\bigg|_{Q^*} \\
&\Leftrightarrow Q_1^*(t,t') = g^2 \sum_{\alpha \in A} n_\alpha \langle \phi(x_\alpha(t))\phi(x_\alpha(t')) \rangle \equiv g^2 C(\tau),
\end{aligned}
\tag{8}
$$

$$
\begin{aligned}
0 \quad &= -\frac{1}{g^2} Q_2^*(t,t') + \sum_{\alpha \in A} \frac{n_\alpha}{Z_\alpha} \frac{\partial Z_\alpha}{\partial Q_1(t,t')}\bigg|_{Q^*} \\
&\Leftrightarrow Q_2^*(t,t') = \frac{g^2}{2} \sum_{\alpha \in A} n_\alpha \langle \tilde{x}_\alpha(t)\tilde{x}_\alpha(t') \rangle = 0,
\end{aligned}
\tag{9}
$$

where $C(\tau)$, with $\tau = t' - t$, represents the average autocorrelation function of the network (as was defined in the main text, *Equation 3*). The second saddle point $Q_2^* = 0$ vanishes due to $\langle \tilde{x}_\alpha(t)\tilde{x}_\alpha(t') \rangle = 0$ as can be immediately extended from *Helias and Dahmen, 2020*; *Sompolinsky and Zippelius, 1982*. The action at the saddle point reduces to the sum of actions for individual, non-interacting units with self-coupling $s_\alpha$. All units are coupled to a common external field $Q_1^*$. Inserting the saddle point values back into *Equation 7*, we obtain $Z^* = \prod_\alpha (Z_\alpha^*)^{N_\alpha}$ where

$$
\begin{aligned}
Z_\alpha^* \sim \quad & \int \mathcal{D}\tilde{x}_\alpha \mathcal{D}x_\alpha \exp \sum_{\alpha \in A} \left( \int dt \tilde{x}_\alpha(t)\big((\partial_t + 1)x_\alpha(t) \right. \\
& \left. -s_\alpha \phi(x_\alpha(t))\big) + \frac{g^2}{2} \int dt dt' \tilde{x}_\alpha(t)C(\tau)\tilde{x}_\alpha(t') \right).
\end{aligned}
\tag{10}
$$

Thus in the large $N$ limit the network dynamics are reduced to those of a number of $A$ units $x_\alpha(t)$, each represents the sub-population with self-couplings $s_\alpha$ and follows dynamics governed by

$$\frac{d}{dt}x_\alpha(t) = -x_\alpha(t) + s_\alpha \phi[x_\alpha(t)] + \eta(t) \tag{11}$$

for all $\alpha \in A$ and where $\eta(t)$ is a Gaussian mean-field with autocorrelation

$$\langle \eta(t)\eta(t') \rangle = g^2 \sum_{\alpha \in A} n_\alpha \langle \phi(x_\alpha(t))\phi(x_\alpha(t')) \rangle. \tag{12}$$

The results above can be immediately extended for the continuous case of self-coupling distribution $P(s)$ yielding:

$$\langle \eta(t)\eta(t') \rangle = g^2 \int p(s)\phi(x(s,t))\phi(x(s,t'))ds \tag{13}$$

with $p(s)$ the density function of the self-couplings distribution in the network and the units dynamics dependent on their respective self-couplings:

$$\frac{d}{dt}x(s,t) = -x(s,t) + s\phi[x(s,t)] + \eta(t) . \tag{14}$$

### An iterative solution

We use an iterative approach to solving the mean-field equations, *Equation 11* and *Equation 12* for a discrete distribution of self-couplings, or *Equation 13* and *Equation 14* for a continuous distribution of self couplings. The approach is adopted from *Stern et al., 2014* and adapted to allow for a consideration of multiple self-couplings. We briefly describe it in what follows. We start by making an initial guess for the mean-field autocorrelation function $C(\tau)$, as defined in *Equation 3*. In its Fourier space, we multiply it by a random angle and $g$ and transform it back to generate an instance of the mean-field $\eta(t)$ (see *Stern et al., 2014* for more details). We create additional $\eta(t)$ instances by repeating the

procedure described above. At least one instance is created per each value $s_\alpha$ drawn from a discrete distribution $P(s)$ of self-couplings with support set $S$, or per each value $s_\alpha$ drawn from $P(s)$ in a case of a continuous distribution. We then solve *Equation 11* (or equivalently *Equation 14* in the case of a continuous distribution) to obtain solutions for $x_\alpha$, one solution for each value of $s_\alpha$. The set of solutions allows us to calculate the set $c_\alpha(t, t') = \langle \phi(x_\alpha(t))\phi(x_\alpha(t')) \rangle$. For a discrete distribution, we then multiply each $c_\alpha$ by its relative weight $n_\alpha$ to compute $C(\tau)$, *Equation 12*. For a continuous distribution, we sum all $c_\alpha$, multiplied by $1/n$, with $n$ their amount, to estimate $C(\tau)$, *Equation 13* (since $s_\alpha$ values were drawn from $P(s)$ each $c_\alpha$ captures approximately $1/n$ of the distribution). We use these sampled mean-field autocorrelations $C(\tau)$ instead of our initial guess to repeat the entire procedure. This leads to obtaining another $C(\tau)$. We iterate until the average across iterations of $C(\tau)$ converges. We note that for the continuous distribution case, we increase the number of drawn $s_\alpha$ values as the iterations progress (starting from very few and ending with many). This allows us to maintain a rapid iterative process and yet receive an accurate solution thanks to the refining of the process with each iteration.

## Fixed points and transition to chaos

Networks with heterogeneous self-couplings exhibit a complex landscape of fixed points $x_\alpha^*$, obtained as the self-consistent solutions to the static version of *Equation 2* and *Equation 3*, namely

$$x_\alpha - s_\alpha \tanh(x_\alpha) = \eta \,, \tag{15}$$

where the mean-field $\eta$ is a Gaussian random variable with zero mean and its variance is given by

$$\langle \eta^2 \rangle = g^2 C \,, \qquad C = \sum_{\alpha \in A} n_\alpha \langle \phi[x_\alpha]^2 \rangle \,. \tag{16}$$

The solution for each unit depends on its respective $s_\alpha$ (*Appendix 1-Figure 1*). If $s_\alpha < 1$ a single interval around zero is available. For $s_\alpha > 1$, for a range of values of $\eta$, $x_\alpha^*$ can take values in 1 of 3 possible intervals. Let us consider a network with two sub-populations with $n_1$ and $n_2 = 1 - n_1$ portions of the units with self-couplings $s_1$ and $s_2$, respectively. In the $(s_1, s_2)$ plane, this model gives rise to a phase diagram with a single chaotic region separating four disconnected stable fixed-point regions (*Figure 2a*). We will first discuss the stable fixed points, which present qualitatively different structures depending on the values of the self-couplings. Within the region of both self-couplings $s_1, s_2 < 1$, the only possibility for a stable fixed point is the trivial solution, with all $x_i = 0$ (*Figure 2a*), where the network activity quickly decays to zero. When at least one self-coupling is greater than one, there are three stable fixed point regions (*Figure 2a*); in these three regions, the network activity starting from random initial conditions unfolds via a long-lived transient period, and then it eventually settles into a stable fixed point. This transient activity with late fixed points is a generalization of the network phase found in *Stern et al., 2014*. When both self-couplings are greater than one ($s_1, s_2 > 1$) the fixed point distribution in each sub-population is bi-modal (*Figure 2a–ii*). When $s_1 > 1$ and $s_2 < 1$, the solutions for the respective sub-populations are localized around bi-modal fixed points and around zero, respectively (*Figure 2a–i*).

However, the available solutions in the latter case are further restricted by stability conditions. In the next Methods section we derive the stability condition by expanding the dynamical *Equation 1* around the fixed point and requiring that all eigenvalues of the corresponding stability matrix are negative. Briefly, the $n_\alpha$ fraction of units with $s_\alpha > 1$ at a stable fixed point are restricted to have support on two disjoint intervals $[x_\alpha^*(s_\alpha) < x_\alpha^-(s_\alpha)] \cup [x_\alpha^*(s_\alpha) > x_\alpha^+(s_\alpha)]$. We refer to this regime as multi-modal, a direct generalization of the stable fixed points regime found in *Stern et al., 2014* for a single self-coupling $s > 1$, characterized by transient dynamics leading to an exponentially large number of stable fixed points. For the $n_\alpha$ portion of units with $s_\alpha < 1$, the stable fixed point is supported by a single interval around zero.

## Stability conditions

To determine the onset of instability, we look for conditions such that at least one eigenvalue develops a positive real part. An eigenvalue of the stability matrix exists at a point $z$ in the complex plane if *Stern et al., 2014*; *Ahmadian et al., 2015*

$$g^2 \sum_{\alpha \in A} n_\alpha \left\langle \frac{\left[1 - \tanh^2(x_\alpha)\right]^2}{\left[z + 1 - s_\alpha \left(1 - \tanh^2(x_\alpha)\right)\right]^2} \right\rangle > 1. \tag{17}$$

The denominator of the expression above is $z$ plus the slope of the curve in **Appendix 1—figure 1 ai and aii**. Hence, a solution whose value $x_\alpha^*$ gives a negative slope (available when $s_\alpha > 1$) leads to a vanishing value of the denominator at some positive $z$ and to a positive eigenvalue and instability. Therefore, the $n_\alpha$ fraction of units with $s_\alpha > 1$ at a stable fixed point are restricted to have support on two disjoint intervals $[x_\alpha^*(s_\alpha) < x_\alpha^-(s_\alpha)] \cup [x_\alpha^*(s_\alpha) > x_\alpha^+(s_\alpha)]$. We refer to this regime as multi-modal, a direct generalization of the stable fixed points regime found in **Stern et al., 2014** for a single self-coupling $s > 1$, characterized by transient dynamics leading to an exponentially large number of stable fixed points. For the $n_\alpha$ portion of units with $s_\alpha < 1$, the stable fixed point is supported by a single interval around zero.

A fixed point solution becomes unstable as soon as an eigenvalue occurs at $z = 0$, obtaining from **Equation 17** the stability condition

$$g^2 \sum_{\alpha \in A} n_\alpha \langle q_\alpha^{-1} \rangle \leq 1 \,, \tag{18}$$

where $q_\alpha = \left[s_\alpha - \cosh^2(x_\alpha)\right]^2$. For $s_\alpha > 1$ the two possible consistent solutions to (**Equation 15**) that may result in a stable fixed point (from the two disjoint intervals in **Appendix 1—figure 1a-i**), contribute differently to $q_\alpha$. Larger $|x_\alpha^*|$ decreases $q_\alpha^{-1}$ (**Appendix 1—figure 1b–i**), thus improving stability. Choices for distributions of $x_\alpha^*$ along the two intervals become more restricted as $g$ increases or $s_\alpha$ decreases, since both render higher values for the stability condition, **Equation 18**, forcing more solutions of $x_i$ to decrease $q_\alpha^{-1}$. This restricts a larger fraction of $x_\alpha^*$ at the fixed points to the one solution with a higher absolute value. At the transition to chaos, a single last and most stable solution exists with all $x_i$ values chosen with their higher absolute value $x_\alpha^*$ (**Appendix 1—figure 1b–i**, light green segments). For those with $s_\alpha < 1$ only one solution is available, obtained by the distribution of $\eta$ through consistency (**Equation 15**) at the fixed point. In this configuration, the most stable solution is exactly transitioning from stability to instability where (**Equation 18**) reaches unity. Hence, the transition from stable fixed points to chaos occurs for a choice of $g$ and $P(s)$ such that solving consistently (**Equation 15**) and (**Equation 16**) leads to saturation of the stability condition (**Equation 18**) at one.

### Universal colored-noise approximation to the Fokker-Planck theory

We consider the special case of a network with two self-couplings where a large sub-population ($N_1 = N - 1$) with $s_1 = 1$ comprises all but one slow probe unit, $x_2$, with large self-coupling $s_2 \gg s_1$. The probe unit obeys the dynamical equation $dx_2/dt = f(x_2) + \eta(t)$, with $f(x) = -x + s_2\phi(x)$. In the large $N$ limit, we can neglect the backreaction of the probe unit on the mean-field and approximate the latter as an external Gaussian colored noise $\eta(t)$ with autocorrelation $g^2 C(\tau) = g^2 \langle \phi[x_1(t)]\phi[x_1(t+\tau)]\rangle$, independent of $x_2$. The noise $\eta(t)$ can be parameterized by its strength, defined as $D = \int_0^\infty d\tau\, C(\tau)$ and its timescale (color) $\tau_1$. For large $s_2$, the dynamics of the probe unit $x_2$ can be captured by a bi-stable chaotic phase whereby its activity is localized around the critical points $x_2 = x^\pm \simeq \pm s_2$ (**Figure 4a–i**) and switches between them at random times. In the regime of colored noise (as we have here, with $\tau_1 \simeq 7.9 \gg 1$), the stationary probability distribution $p(x)$ (for $x \equiv x_2$, **Figure 4a–ii and b**) satisfies the unified colored noise approximation to the Fokker Planck equation (**Hänggi and Jung, 1995**; **Jung and Hänggi, 1987**):

$$p(x) = Z^{-1}|h(x)|\exp\left[-U_{eff}(x)/D\right] \,, \tag{19}$$

where $Z$ is a normalization constant, $h(x) \equiv 1 - \tau_1 f'(x)$, and the effective potential $U_{eff}(x) = -\int^x f(y)h(y)dy$ is therefore:

$$U_{eff} = \frac{x^2}{2} - s_2 \log \cosh(x) + \frac{\tau_1}{2}f(x)^2 - U_{min} \,. \tag{20}$$

The distribution $p(x)$ has support in the region $h(x) > 0$ comprising two disjoint intervals $|x| > x_c$ where $\tanh(x_c)^2 = 1 - \dfrac{1 + \tau_1}{\tau_1 s_2}$ (*Figure 4b*). Moreover, the probability distribution is concentrated around the two minima $x^\pm \simeq \pm s_2$ of $U_{eff}$. The new UCNA-based term $\dfrac{\tau_1}{2} f'(x)^2$ dominates the effective potential. The main effect of the strong color $\tau_1 \gg 1$ is to sharply decrease the variance of the distribution around the minima $x^\pm$. This is evident from comparing the colored noise with white noise, when the latter is driving the same bi-stable probe $dx_2/dt = -x_2 + s_2\phi(x_2) + \xi(t)$, where $\xi(t)$ is a white noise with an equivalent strength to the colored noise, *Figure 4iv-vi*. The naive potential for the white noise case $U = x^2/2 - s_2 \log \cosh(x)$ is obtained from *Equation 19* by sending $\tau_1 \to 0$ in the prefactor $h$ and in potential $U_{eff}$. It results in wider activity distribution compared to our network generated colored noise, in agreement with the simulations, *Figure 4b*.

In our colored-noise network, the probe unit's temporal dynamics are captured by the mean first passage time $\langle T \rangle$ for the escape out of the potential well:

$$\langle T \rangle = \int_{-s_2}^{-x_c} \frac{dx}{D} \frac{h(x)^2}{p(x)} \int_{-\infty}^{x} p(y)dy , \qquad (21)$$

where the upper limit $x_c$ in the outer integral is the edge of the support of $p(x)$. In the small $D$ approximation, we can evaluate the integrals by steepest descent. The inner integrand $p(x)$ is peaked at the minimum $x^- = -s_2$ of the effective potential, yielding

$$\int_{-\infty}^{x} p(y)dy = Z^{-1} \sqrt{\frac{2\pi}{D} U_{eff}''(x^-)} \exp\left(-U_{eff}(x^-)/D\right) .$$

The outer integrand can be rewritten as $\psi(x) = \exp\dfrac{\rho(x)}{D}$, where $\rho(x) = U_{eff}(x) + D\ln|h(x)|$ peaks at $-x_f$ with $\tanh(x_f)^2 \simeq 1 - 1/2s_2$. The mean first passage time can thus be estimated as

$$\langle T \rangle \simeq 2\pi \sqrt{U_{eff}''(x^-)\rho''(x_f)} \exp\left(\frac{\Delta}{D}\right) , \qquad (22)$$

where $\Delta = \rho(x_f) - U_{eff}(x^-)$ and its asymptotic scaling for large $s_2$ leads to *Equation 4*. We validated the UCNA approach to calculate the mean first passage time by estimating the distribution of escape points $x_{esc}$ from one well to the other well, which was found to lie predominantly within the support $x > |x_c|$ of the stationary probability distribution $p(x)$. Only a fraction of activity in the simulations $(1.8+/-0.4) * 10^{-3}$ (mean±SD over 10 probe units run with parameters as in *Figure 4b*) entered the forbidden region (see *Figure 4—figure supplement 1* for details ).

## A comparison with white noise

To test the impact of the input generated by the network (or equivalently as mimicked by the colored noise), we replaced this input (*Equation 1*, most rhs term) with white noise. The probe unit $x$ in the white noise case is, therefore, following the dynamical equation:

$$\frac{dx}{dt} = -x + s\phi(x) + g\sqrt{D}\eta(t) \qquad (23)$$

with $\eta(t)$ taken from a normal distribution, $\phi \equiv \tanh$ and $s, g$, and $D$ are constants receiving their values according to the probe unit dynamics driven by the network case; specifically these constants are the probe unit self-coupling strength, the original network gain, and strength (the integral under the autocorrelation function of the network input to the probe). Simulation results of the probe dynamics are in *Figure 4aii*, along with its distribution, *Figure 4b* (light green area, parameters' values for $s, g$, and $D$ are specified in the caption). To estimate the probability (*Equation 19*) and the potential (*Equation 20*) in this case, and since $\eta$ here is white noise, we substitute $\tau_1 = 0$ as no correlation in the input exists. Similarly, we calculate the probe's approximated mean first passage time when driven by white noise (*Equation 4*). The result (*Figure 4c* light green dashed line) estimates the simulations well (*Figure 4c* green line). Note that since $\log < T >$ depends on $\tau_1$ linearly, its exponent, the mean first passage time, depends on $\tau_1$ exponentially. Hence, the importance of the 'color' (correlations) in the network input in generating long timescales and the failure of these long

**Table 1.** Parameters for the clustered network used in the simulations.

**Model parameters for clustered network simulations**

| Parameter | Description | Value |
|---|---|---|
| $J_{EE}$ | E-to-E synaptic weights | $0.9/\sqrt{N}$ [mV] |
| $J_{IE}$ | E-to-I synaptic weights | $0.9/\sqrt{N}$ [mV] |
| $J_{EI}$ | I-to-E synaptic weights | $2.7/\sqrt{N}$ [mV] |
| $J_{II}$ | I-to-I synaptic weights | $5.4/\sqrt{N}$ [mV] |
| $J_{E0}$ | E-to-E synaptic weights | $3.7/\sqrt{N}$ [mV] |
| $J_{I0}$ | I-to-I synaptic weights | $3.3/\sqrt{N}$ [mV] |
| $J_{EE}^{+}$ | Potentiated intra-assembly E-to-E weight factor | $14\sqrt{N/2000}$ |
| $J_{II}^{+}$ | Potentiated intra-assembly I-to-I weight factor | $5\sqrt{N/2000}$ |
| $g_{EI}$ | Potentiation parameter for intra-assembly I-to-E weights | $10\sqrt{N/2000}$ |
| $g_{IE}$ | Potentiation parameter for intra-assembly E-to-I weights | $8\sqrt{N/2000}$ |
| $r_{ext}$ | Average baseline afferent rate to E and I neurons | 5 [spks/s] |
| $V_E^{thr}$ | E-neuron threshold potential | 1.43 [mV] |
| $V_I^{thr}$ | I-neuron threshold potential | 0.74 [mV] |
| $V^{reset}$ | E- and I-neuron reset potential | 0 [mV] |
| $\tau_m$ | E- and I-neuron membrane time constant | 20 [ms] |
| $\tau_{refr}$ | E- and I-neuron absolute refractory period | 5 [ms] |
| $\tau_s$ | E- and I-neuron synaptic time constant | 5 [ms] |

timescales to materialize when the 'color' is removed (as in this particular white noise-driven probe case, which replicates the assemblies endowed network model except for its generated correlated input drive).

## Spiking network model

### Network architecture

We simulated a recurrent network of $N$ excitatory (E) and inhibitory (I) spiking neurons (for $N = 2000, 5000, 10000$) with relative fractions $n_E = 80\%$ and $n_I = 20\%$ and connection probabilities $p_{EE} = 0.2$ and $p_{EI} = p_{IE} = p_{II} = 0.5$ (**Figure 5**). Non-zero synaptic weights from pre-synaptic neuron $j$ to post-synaptic neuron $i$ were $J_{ij}$, whose values only depended on the two neurons types $i, j \in \{\alpha, \beta\}$ for $\alpha, \beta = E, I$. Neurons were arranged in $p$ cell-type specific assemblies. E assemblies had heterogeneous sizes drawn from a uniform distribution with a mean of $N_E^{clust} = 60 + N/100$ E-neurons and 30% standard deviation. The number of assemblies was determined as $p = \text{round}(n_E N(1 - n_{bgr})/N_E^{clust})$, where $n_{bgr} = 0.1$ is the fraction of background neurons in each population, i.e., not belonging to any assembly. I assemblies were paired with E assemblies and the size of each I assembly was matched to the corresponding E assembly with a proportionality factor $n_I/n_E = 1/4$. Neurons belonging to the same assembly had potentiated intra-assembly weights by a factor $J_{\alpha\beta}^{+}$, while those belonging to different assemblies had depressed inter-assembly weights by a factor $J_{\alpha\beta}^{-}$, where: $J_{EI}^{+} = p/(1 + (p - 1)/g_{EI})$, $J_{IE}^{+} = p/(1 + (p - 1)/g_{IE})$, $J_{EI}^{-} = J_{EI}^{+}/g_{EI}$, $J_{IE}^{-} = J_{IE}^{+}/g_{IE}$ and $J_{\alpha\alpha}^{-} = 1 - \gamma(J_{\alpha\alpha}^{+} - 1)$ for $\alpha = E, I$, with $\gamma = f(2 - f(p + 1))^{-1}$. $f = (1 - n_{bgr})/p$ is the fraction of E neurons in each assembly. Parameter values are in **Table 1**.

## Single neuron dynamics

Single neuron dynamics. We simulated current-based leaky-integrate-and-fire (LIF) neurons, with membrane potential $V$ and dynamical equation

$$\frac{dV}{dt} = -\frac{V}{\tau_m} + I_{rec} + I_{ext} \, ,$$

where $\tau_m$ is the membrane time constant. Input currents included a contribution $I_{rec}$ from the other recurrently connected neurons and a constant external current $I_{ext} = N_{ext}J_{\alpha 0}r_{ext}$ (units of mV s$^{-1}$), for $\alpha = E, I$, representing afferent inputs from other brain areas and $N_{ext} = n_E N p_{EE}$. When the membrane potential $V$ hits the threshold $V_\alpha^{thr}$ (for $\alpha = E, I$), a spike is emitted and $V$ is held at the reset value $V^{reset}$ for a refractory period $\tau_{refr}$. We chose the thresholds so that the homogeneous network (i.e. where all $J_{\alpha\beta}^\pm = 1$) was in a balanced state with average spiking activity at rates $(r_E, r_I) = (2, 5)$ spks/s. The post-synaptic currents evolved according to

$$\tau_{syn}\frac{dI_{rec}}{dt} = -I_{rec} + \sum_{j=1}^{N} J_{ij} \sum_k \delta(t - t_k) \, ,$$

where $\tau_s$ is the synaptic time constant, $J_{ij}$ are the recurrent couplings and $t_k$ is the time of the k-th spike from the j-th presynaptic neuron. Parameter values are in **Table 1**.

## Self-couplings from mean-field theory

We can estimate the E-assembly self-couplings in this model using mean-field methods (**Amit and Brunel, 1997**; **Wyrick and Mazzucato, 2021**). This method allows obtaining, self-consistently, the fixed point values of the firing rates $r_l^E, r_l^I$ in the l-th assembly ($l = 1, \ldots, p$) via the equation

$$r_l^\alpha = F_\alpha[\mu_l^\alpha(\mathbf{r}), \sigma_l^\alpha(\mathbf{r})] \, , \tag{24}$$

where $\mathbf{r} = (r_1^E, \ldots, r_p^E, r_1^I, \ldots, r_p^I)$ is the leaky-integrate-and-fire current-to-rate transfer function for each $\alpha = E, I$ population

$$F_\alpha(\mu_l^\alpha, \sigma_l^\alpha) = \left( \tau_{refr} + \tau_m^\alpha \sqrt{\pi} \int_H^\Theta du\, e^{u^2}[1 + \mathrm{erf}(u)] \right)^{-1} \, , \tag{25}$$

where $H_l = (V^{reset} - \mu_l^\alpha)/\sigma\alpha_l + ak$ and $\Theta = (V_l^{thr} - \mu_l^\alpha)/\sigma_l^\alpha + ak$ and $a = |\zeta(1/2)|/\sqrt{2}$ are terms accounting for the synaptic dynamics (**Fourcaud and Brunel, 2002**). The infinitesimal means $\mu_l^E, \mu_l^I$ and variances $(\sigma_l^E)^2, (\sigma_l^I)^2$ of the network populations comprising E and I assemblies (for $l = 1, \ldots, p$ assemblies) are themselves functions of the firing rates, thus leading to self-consistent equations for the fixed points (for more details see **Wyrick and Mazzucato, 2021**). The infinitesimal mean $\mu_1^E$ of the postsynaptic input to a neuron in a representative E assembly in focus is

$$\begin{aligned}
\tau_E^{-1}\mu_1^E = &Nn_E p_{EE}J_{EE}\Big[J_{EE}^+ f_1^E r_1^E \\
&+ J_{EE}^- \sum_{l=2}^p f_l^E r_l^E + n_{bg}r_{bg}^E\Big] + N_{ext}J_{E0}r_{ext} \\
&- Nn_I p_{EI}J_{EI}\Big[f_1^d J_{EI}^+ r_1^I + J_{EI}^- \sum_{l=2}^p f_l^I r_l^I + n_{bg}r_{bg}^I\Big] \, ,
\end{aligned} \tag{26}$$

where $r_1^E$ is the firing rate of the E assembly in focus and $r_1^I$ is the firing rate of its paired I assembly; $r_l^E, r_l^I$, for $l = 2, \ldots, p$ are the firing rates of the other E and I assemblies; $r_{bg}^E, r_{bg}^I$ are the firing rates of the background (unclustered) populations. $f_i^E, f_i^I$ represent the fraction of E and I neurons in each assembly, which are drawn from a uniform distribution (see above). The first line in **Equation 26** represents the contribution to the input current coming from neurons within the same E assembly, or, in other words, the self-coupling of the assembly in focus. We can thus recast the first term in the input current as $s_1 r_1^E$ where $s_1 = Nn_E p_{EE}J_{EE}J_{EE}^+ f_1^E$. The number of neurons in the assembly is given by $N_1 = Nn_E f_1^E$, and the average E-to-E synaptic coupling is $\bar{J}^{(in)} = p_{EE}J_{EE}J_{EE}^+$, from which we obtain $s_1 = N_1 \bar{J}_{EE}^{(in)}$, which is the expression we used in **Figure 5**. We can thus recast **Equation 26** as

$$\tau_E^{-1}\mu_1^E = \quad s_1^E r_1^E - s_1^I r_1^I + \sum_{l=2}^p (\hat{J}_{1l}^{EE} r_l^E - \hat{J}_{1l}^{EI} r_l^I)$$
$$+\hat{J}^{bg,E} r_{bg}^E - \hat{J}^{bg,I} r_{bg}^I + \hat{J}^{ext} r_{ext} \, , \tag{27}$$

where $\hat{J}$ represent effective synaptic couplings which depend on the underlying spiking network parameters in *Equation 26*. In the spiking model, the self-couplings have units of [mV]. The first line in *Equation 27* exhibits the same functional form as the rate model in *Equation 1*, if we identify each rate unit as a functional assembly with a corresponding self-coupling. A crucial simplification occurring in the rate model *Equation 1* is the absence of cell-type specific connectivity and the corresponding difference in the statistics of the distribution of the effective couplings $\hat{J}$, whose mean is zero in *Equation 1* but non-zero in *Equation 27*. If we interpret the current $x$ in the rate model as a membrane potential with units of [mV] (see *Miller and Fumarola, 2012*), and the current-to-rate function $\phi(x) = \tanh(x)$ as a normalized (min-maxed) firing rate fluctuation around baseline (see *Ahmadian et al., 2015*), then the self-coupling in the rate model exhibits units of [mV] as in the spiking model. However, direct numerical comparison of the self-couplings between the two models is hampered by the fact that the spiking model is a balanced network, where E and I contributions to the total input current are large and cancel to leading order (*Wyrick and Mazzucato, 2021*), whereas the rate network does not operate in the balanced regime.

## Model selection for timescale fit

Model selection for timescale fit The degree of the polynomial that best fits the dependence of the logarithm of the assembly timescales on the assembly self-couplings was estimated using cross-validation (inset in *Figure 5b*), according to the following steps. (1) The dataset was split into a training set and a test set of the same size. (2) Polynomial functions with increasing degrees were fit to the training set. (3) The mean-squared error of the test set was obtained from the corresponding fit. (4). A minimum was achieved for a polynomial of degree 2. All logarithms, abbreviated as 'log' in the main text, is in the natural base $e$.

# Acknowledgements

We would like to thank J Wallis and S Cavanagh for sharing their previously published data *Cavanagh et al., 2016*; and G Mongillo and G La Camera for discussions. LM was supported by National Institute of Neurological Disorders and Stroke grant R01-NS118461 and by the National Institute on Drug Abuse grant R01-DA055439 (CRCNS). MS was supported by The Hebrew University of Jerusalem 'Emergency response to COVID-19' grant and hosted by the Theoretical Sciences Visiting Program at the Okinawa Institute of Science and Technology while completing this study.

# Additional information

## Funding

| Funder | Grant reference number | Author |
| --- | --- | --- |
| National Institute of Neurological Disorders and Stroke | R01-NS118461 | Luca Mazzucato |
| National Institute on Drug Abuse | R01-DA055439 | Luca Mazzucato |

The funders had no role in study design, data collection and interpretation, or the decision to submit the work for publication.

## Author contributions

Merav Stern, Conceptualization, Software, Formal analysis, Validation, Investigation, Visualization, Methodology, Writing – original draft, Writing – review and editing; Nicolae Istrate, Software, Formal analysis, Validation, Investigation, Visualization, Methodology, Writing – original draft, Writing – review and editing; Luca Mazzucato, Conceptualization, Software, Formal analysis, Supervision, Funding

acquisition, Validation, Investigation, Visualization, Methodology, Writing – original draft, Project administration, Writing – review and editing

## Author ORCIDs
Merav Stern  http://orcid.org/0000-0001-7744-4855
Luca Mazzucato  https://orcid.org/0000-0002-8525-7539

## Decision letter and Author response
Decision letter https://doi.org/10.7554/eLife.86552.sa1
Author response https://doi.org/10.7554/eLife.86552.sa2

## Additional files

### Supplementary files
• MDAR checklist

### Data availability
https://github.com/nistrate/multipleTimescalesRNN. (Copy archive at *Istrate, 2022*).

The following previously published dataset was used:

| Author(s) | Year | Dataset title | Dataset URL | Database and Identifier |
|---|---|---|---|---|
| Cavanagh S, Wallis J, Kennerley S, Hunt L | 2020 | Data from: Autocorrelation structure at rest predicts value correlates of single neurons during reward-guided choice | https://doi.org/10.5061/dryad.5b331 | Dryad Digital Repository, 10.5061/dryad.5b331 |

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

## Appendix 1

### Dynamical regions of networks with identical self-couplings

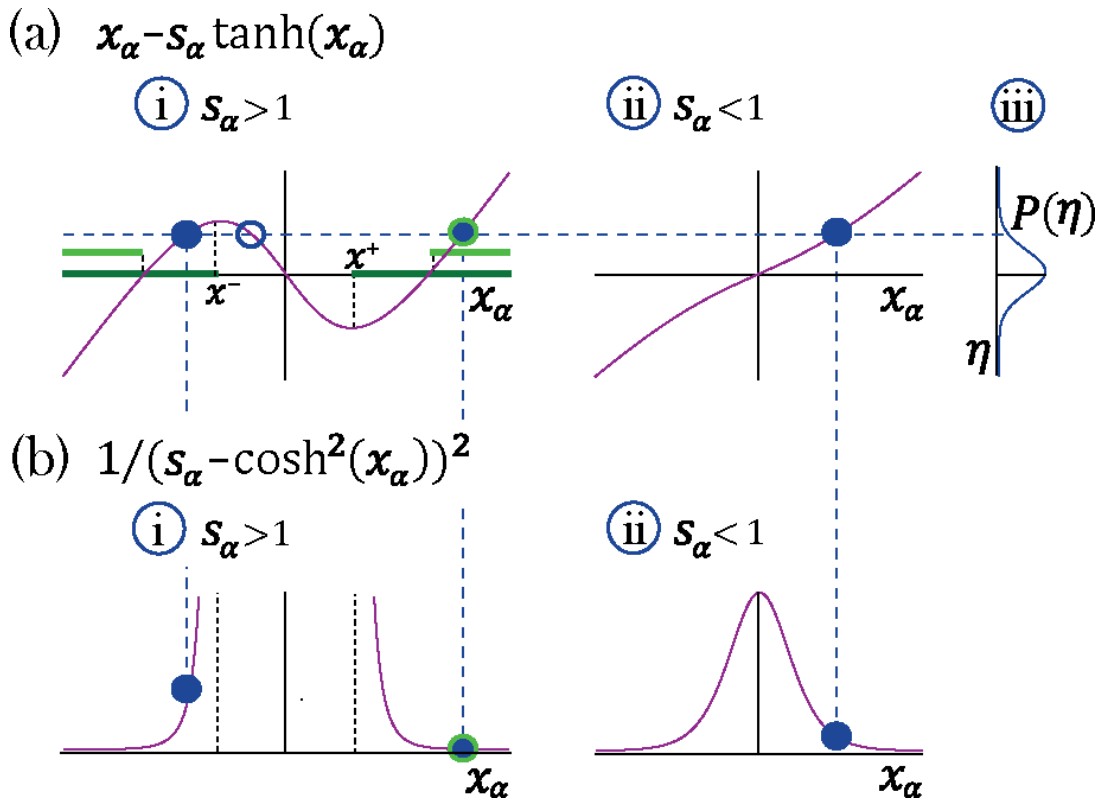

(a) $x_\alpha - s_\alpha \tanh(x_\alpha)$

(i) $s_\alpha > 1$    (ii) $s_\alpha < 1$    (iii)

(b) $1/(s_\alpha - \cosh^2(x_\alpha))^2$

(i) $s_\alpha > 1$    (ii) $s_\alpha < 1$

**Appendix 1—figure 1.** Transition to chaos with multiple self-couplings: Fixed point solutions and stability. (a–i) The fixed point curve $x_\alpha - s_\alpha \tanh x_\alpha$, from **Equation 15**, for $s_\alpha > 1$. Stable solutions are allowed within the dark green region. (b–i) The shape of a unit's contribution to stability $q^{-1} = (s_\alpha - \cosh x_\alpha^2)^{-2}$, from **Equation 18**. Stable solutions of $x_\alpha - s_\alpha \tanh x_\alpha = \eta$, filled blue circles in (a–i), with different |x| values contribute differently to stability. At the edge of chaos only a fixed point configuration with all units contributing most to stability (minimal $q^{-1}$) is stable, light green region in (a–i).(a–ii) The curve $x_\alpha - s_\alpha \tanh x_\alpha$ for $s_\alpha < 1$. (a–iii) A possible distribution of the Gaussian mean-field $\eta$. A representative fixed point solution is illustrated by the dashed blue line: for $s_\alpha < 1$ a single solution exists for all values of $\eta$, (filled blue circle in a-ii); For $s_\alpha > 1$ multiple solutions exist (a–i) for some values of $\eta$; some of them lead to instability (empty blue circle in a-i). The other two solutions may lead to stability (filled blue circles in a-ii), although only one of them will remain stable at the edge of chaos (encircled with green line in a-i).

It is constructive to quickly survey the results of **Stern et al., 2014** who studied the special case of including a single value self-coupling $s$ for all assemblies in the network, $P(s_i) = \delta_{s,s_i}$. In this case, the dynamics of all units in the network follow:

$$\frac{dx_i}{dt} = -x_i + s \tanh(x_i) + g \sum_{i=1}^{N} J\phi(x_j), \tag{28}$$

Two variables determine the network dynamics, the network gain $g$ and the self-coupling value $s$. The network gain $g$ defines the strength of the network impact on its units. It brings the network into chaotic activity, without self-coupling ($s = 0$), for values $g > 1$ (**Sompolinsky et al., 1988**). The self-coupling $s$ generates bi-stability. Without network impact ($g = 0$) the dynamical **Equation 28** for each unit becomes

$$\frac{dx_i}{dt} = -x_i + s \tanh(x_i), \tag{29}$$

which has two stable solutions for $s > 1$ (**Appendix 1—figure 2a**), both at $x \neq 0$. For $s < 1$ (**Appendix 1—figure 2b**), a single stable solution exists at $x = 0$.

When small values of network gain $g$ are introduced to the network dynamics, **Equation 28**, with identical bi-stable units ($s > 1$), each unit solution jitters around one of its two possible fixed points. After an irregular activity, the network settles into a stable fixed point. This generates a region of transient irregular activity with stable fixed points (**Appendix 1—figure 2c**). As $g$ increases and $s$ decreases, different possible fixed point configurations lose their stability (as a result, the typical time spent in the transient activity increases). When the critical line $s_c \approx 1 + 0.157 \ln(0.443g + 1)$ is crossed, no fixed point remains stable and the network activity becomes chaotic (**Stern et al., 2014**). The 'last' stable fixed point at the transition line has a unique configuration with all unit values located farthest from $x = 0$ (**Appendix 1—figure2a**, light green lines). Additional decrease of $s$ and $g$ leads to a region where any initial activity of the network decays and the trivial solution ($x_i = 0$ for all $i$) is stable (**Appendix 1—figure 2c**)

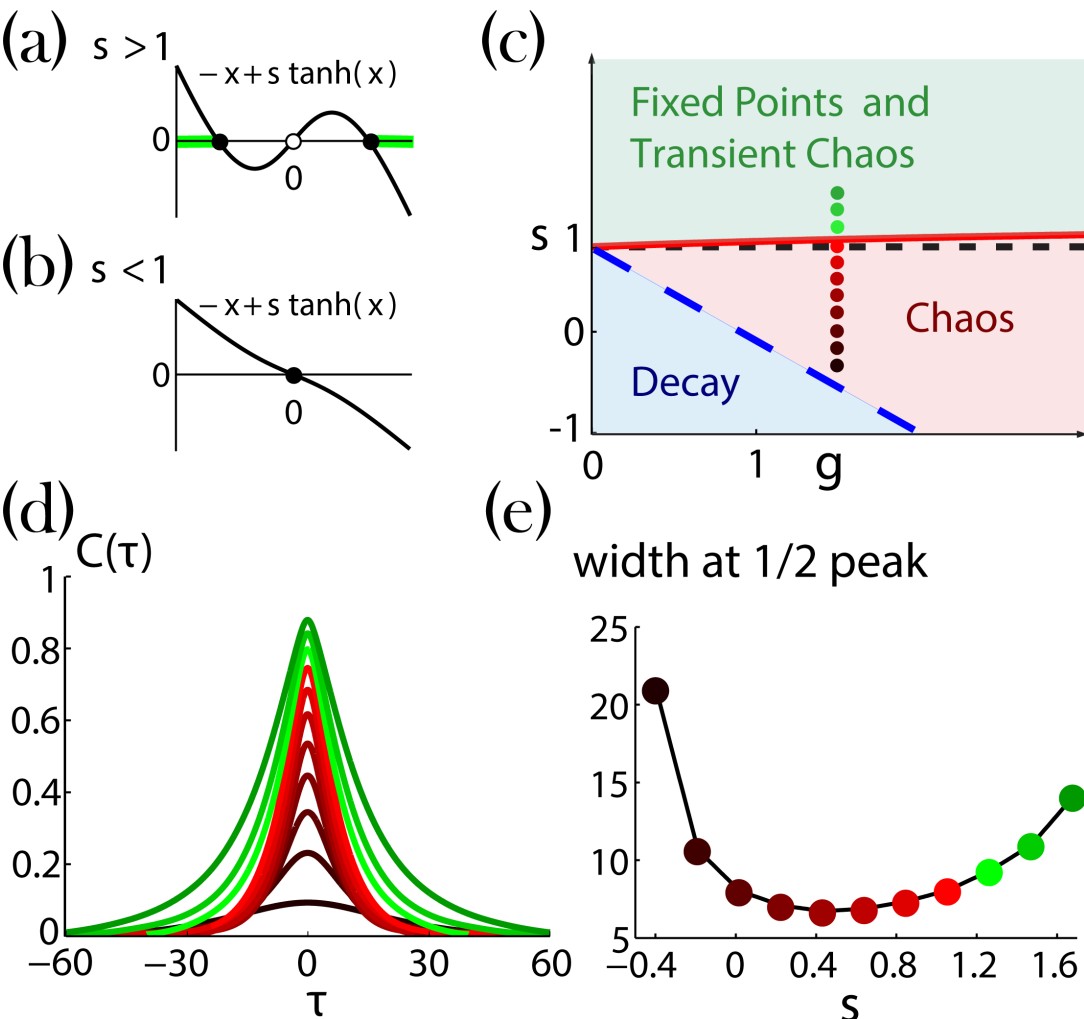

**Appendix 1—figure 2.** Network dynamics with identical self-couplings, adopted from **Stern et al., 2014**. (a,b) Graphical solutions to **Equation 29**. (a) For $s > 1$ there are two stable non-zero solutions (full black circles) and an unstable solution at zero (open black circle). The green background over the $x$ axis denotes the regions of allowed activity values at the stable fixed point on the transition line to chaos (solid red curve in (c)).(b) For $s < 1$ there is a single stable solution (full black circle) at zero. (c) Regions of the network dynamics over a range of $s$ and $g$ values. Below the long dashed blue line, any initial activity in the network decays to zero. Above the solid red curve, the network exhibits transient irregular activity that eventually settles into one out of a number of possible nonzero stable fixed points. In the region between these two curves, the network activity is chaotic. Colored circles denote, according to their locations on the phase diagram and with respect to their colors, the values of $s$ (ranging

*Appendix 1—figure 2 continued*

from 1.6 and decreasing with steps of 0.2) and $g = 1.5$, used for the autocorrelation functions $C(\tau)$ in (**d**; corrected version of Figure 4a in *Stern et al., 2014*). (**e**) Widths at half peak (values of $\tau$'s in the main text notation) of the autocorrelation functions in (**d**).

## Appendix 2

### Metastable dynamics in the spiking network model

The spiking network model with E/I clusters is based on a weight matrix $W$ with four E/I submatrices, each one exhibiting diagonal blocks representing potentiated intra-cluster synaptic weights (see *Appendix 2—figure 1a, b*). We can approximate the full $N \times N$ weight matrix with a mean-field reduction to a set of $2(p+1)$ mean-field variables $r_i^{MF}$, representing the $p$ E and I clusters and the two unclustered background E and I populations. In order to gain an intuitive understanding of the population dynamics encoded by $J^{MF}$, we follow the approach in *Murphy and Miller, 2009*; *Schaub et al., 2015* and can consider an auxiliary linear system

$$\tau_m \frac{dr_i^{MF}}{dt} = -r_i^{MF} + \sum_{j=1}^{2p+2} J_{ij}^{MF} r_j^{MF} , \tag{30}$$

which was shown to capture the transition from asynchronous irregular activity to metastable attractor dynamics in *Schaub et al., 2015*. A Schur decomposition of $J^{MF} = VTV^T$ gives an upper triangular matrix $T$, whose Schur eigenvectors (columns of $V$) represent independent dynamical modes (i.e. an orthonormal basis; *Appendix 2—figure 1b*). The Schur eigenvalues (diagonal values in the Schur matrix) correspond to the real part of the eigenvalues of the original matrix $J^{MF}$. The Schur eigenvectors associated with the large positive eigenvalues approximately correspond to E/I cluster pairs; larger eigenvalues correspond to clusters of increasingly larger size (*Appendix 2— figure 1b*).

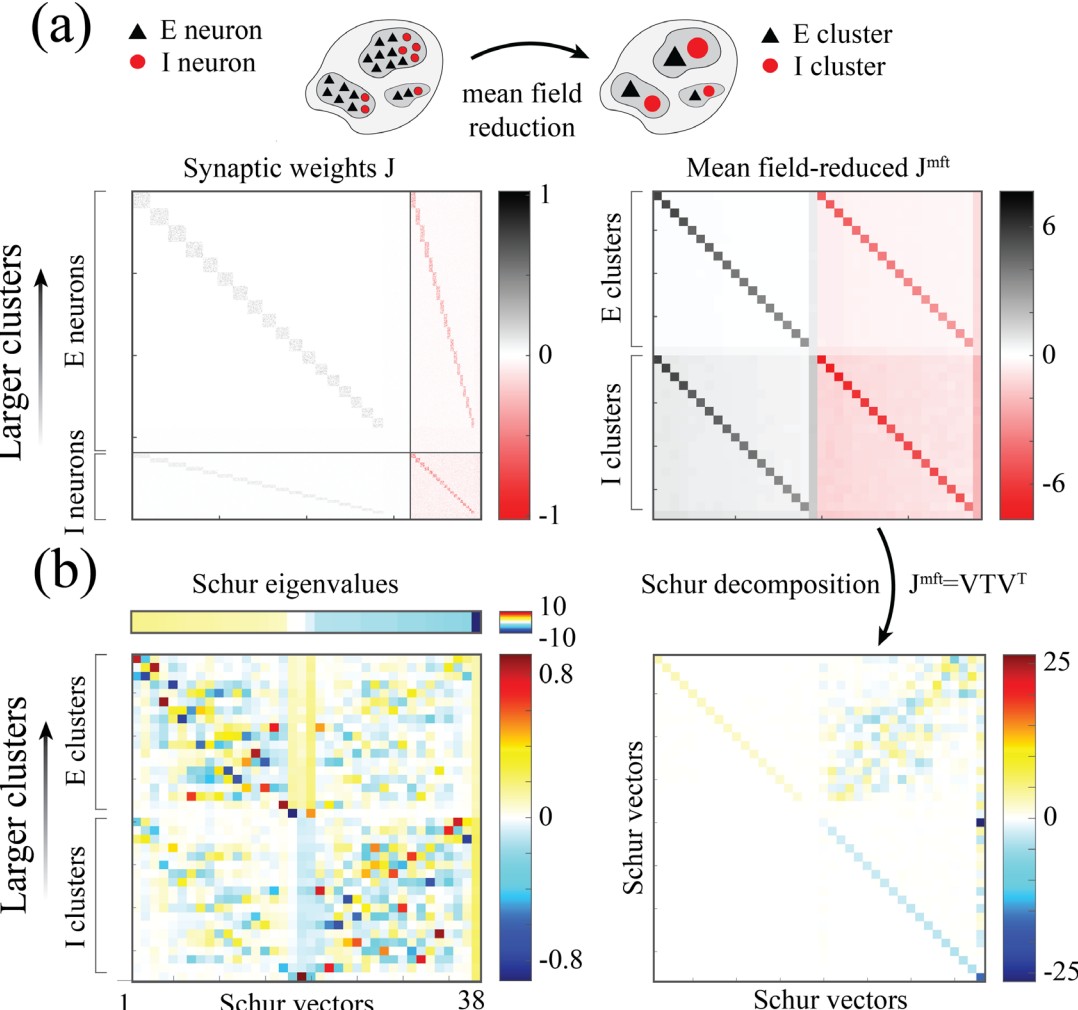

**Appendix 2—figure 1.** Synaptic weight matrices of clustered spiking networks. Synaptic weight matrices of a clustered spiking networks. (**a**) Left: Synaptic weight matrix $J$ of a clustered spiking network with $N = 2000$ neurons, exhibiting the E/I four submatrices, where diagonal blocks within each submatrix reveal the E/I clustered structure (larger to smaller clusters, top to bottom in each submatrix). Right: $2(p + 1)$-dimensional mean-field-reduced synaptic weight matrix $J^{MF}$ corresponding to the full matrix on the left. Populations are ordered as follows: $p + 1$ excitatory clusters (larger to smaller, top to bottom), background excitatory population, $p + 1$ inhibitory clusters, background inhibitory population. For each pair of populations, the value in $J^{MF}$ is obtained from the corresponding block in $J$ by summing its column values (presynaptic inputs) and averaging over the rows (postsynaptic neurons belonging to the population). (**b**) Right: Schur decomposition of $J^{MF} = VTV^T$ into an upper triangular matrix $T$. Left: Schur eigenvectors (columns of $V$, bottom) sorted from larger to smaller Schur eigenvalue (top). Larger Schur eigenvalues are associated with eigenvectors whose loadings are on larger E/I cluster pairs.

Clustered networks generate metastable attractor dynamics where coupled E/I cluster pairs switch between periods of low and high firing rate activity, yielding a bimodal firing rate distribution (*Appendix 2—figure 2a, b*, see *Litwin-Kumar and Doiron, 2012*; *Deco and Hugues, 2012*; *Mazzucato et al., 2015*; *Mazzucato et al., 2019*; *Mazzucato et al., 2016*; *Wyrick and Mazzucato, 2021*; *Schaub et al., 2015*). A more detailed analysis revealed that the network activity explores metastable attractors with varying number of simultaneously active clusters (from one to four in the representative network with $N = 2000$ neurons in *Appendix 2—figure 2c*), yielding a complex attractor landscape (*Mazzucato et al., 2015*; *Wyrick and Mazzucato, 2021*). The firing rate of neurons in active clusters is inversely proportional to the number of simultaneously active clusters. Therefore, the activity of single neurons is not simply bistable, but rather multistable, with several levels of firing rates attainable depending on which attractor the network is expressing at any given time (one inactive and four active levels in this representative case). This single-neuron multistability

property is a biologically plausible effect observed in cortical activity (*Mazzucato et al., 2015*; *Recanatesi et al., 2022*), however, it is not present in the rate network, where neurons are just bistable (*Figure 4*). In the spiking model, clusters are uncorrelated (0.01±0.12, mean±S.D. across 20 networks; *Appendix 2—figure 2c*), similarly to neurons in the rate network.

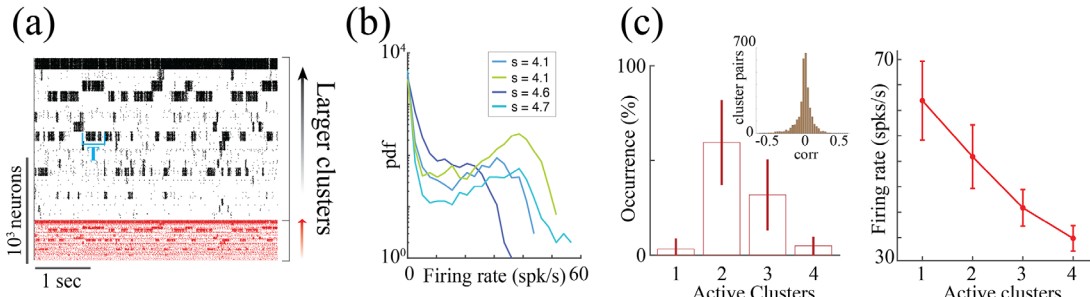

**Appendix 2—figure 2.** Metastable attractor dynamics in clustered spiking networks. Metastable attractor dynamics in clustered spiking networks. (**a**) Metastable activity from all clustered neurons in a representative trial of a network with $N = 2000$ neurons (larger to smaller clusters, top to bottom; action potentials from excitatory and inhibitory neurons, black and red, respectively). (**b**) Firing rate distributions of four representative neurons from the network in (**a**) exhibit bimodal distributions (colors represent clusters with different self-couplings; spike counts estimated in 100ms bins over 20 trials of 200 seconds duration). (**c**) Left: Metastable attractor dynamics unfolds through different network configurations with a range of simultaneously active clusters from one to four (average occurrence of each configuration across 20 networks as fraction of total simulation time). Inset: Metastable dynamics are uncorrelated across clusters (distribution of pairwise Pearson correlations between clusters' firing rates, 0.01±0.12, mean±SD; 20 networks). Right: The firing rate of neurons in active clusters depend on how many clusters are simultaneously active, with higher firing rates in configurations with less co-active clusters (mean and SD across 20 networks).

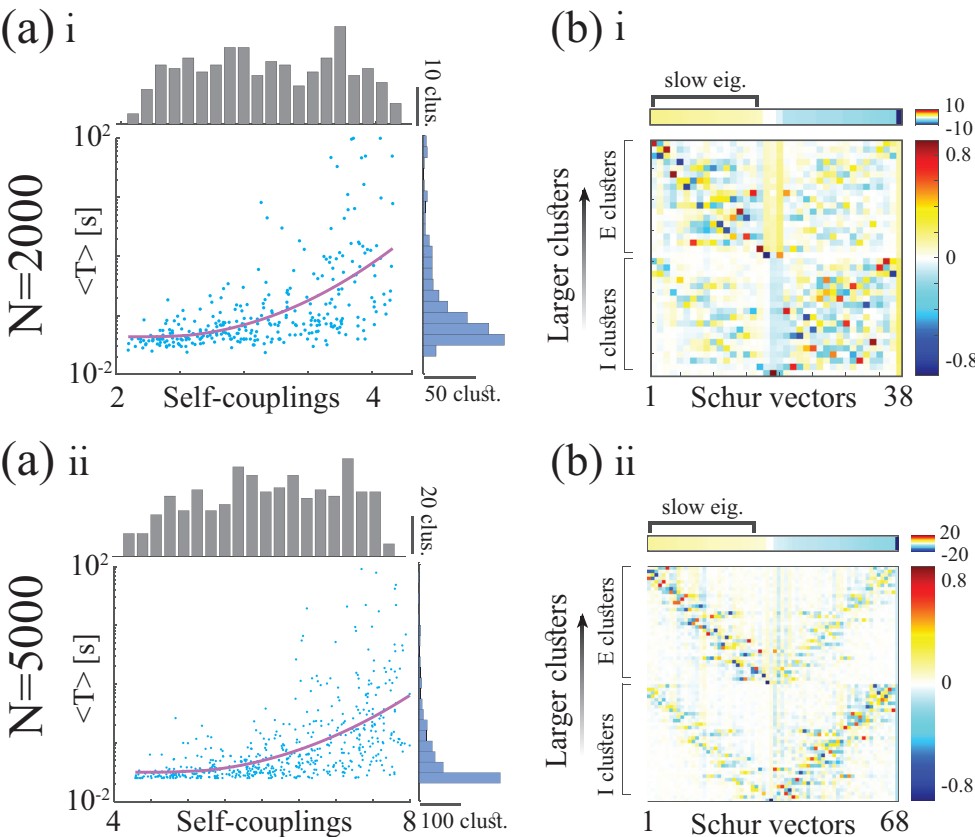

**Appendix 2—figure 3.** Metastable activity and timescale distribution in networks of $N = 2000, 5000$ neurons (top and bottom rows, respectively). Panels (**a**) and (**b**) have the same notations as *Figure 5b*, and *Figure 5d*, respectively. The fits of average activation time $T$ vs. self-coupling $\log(T) = a_2 s_E^2 + a_1 s_E + a_0$ yielded $a_2 = 0.44, a_1 = 2.06, a_0 = 1.04$ for $N = 2000$; and $a_2 = 0.093, a_1 = -0.88, a_0 = 0.45$ for $N = 5000$.

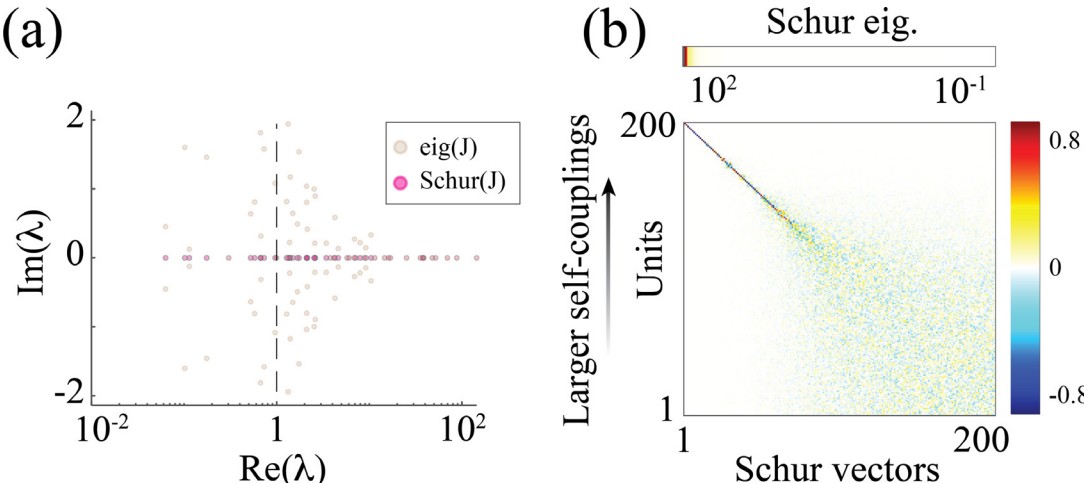

**Appendix 2—figure 4.** Chaotic rate network of $N = 200$ units (self-couplings drawn from a lognormal distribution with parameters $\mu = 0.2, \sigma = 2$). (**a**) Eigenvalue distribution (brown) and Schur eigenvalues (pink) of the synaptic weight matrix $J$. (**b**) The Schur eigenvectors of $J$ corresponding to large positive Schur eigenvalues have loadings localized on units with larger self-couplings, with a similar structure to the Schur eigenvectors in the spiking networks in *Appendix 2-figure 3* .

## Appendix 3

### RNN with heterogeneous time constants

Our recurrent neural network model in *Equation 1*, assumes that all units share the same time constant, $\theta = 1$ ms, which measures the rate of change of a neuron's membrane potential. We examined whether a network of units with heterogeneous time constants could give rise to multiple timescales of dynamics. We simulated the model from *Equation 1* with no self-coupling term, $s_i = 0$, with neuron-specific time constant $\theta_i$:

$$\theta_i \frac{d}{dt} x_i(t) = -x_i(t) + g \sum_j J_{ij} \phi \left[ x_j(t) \right].$$

(31)

Following the same strategy as in *Figure 2*, we consider the scenario when our network contains two equal-size populations of neurons ($N_1 = N_2$) with different time constants $\theta_1 \neq \theta_2$. We quantified each unit's timescales, $\tau_i$, as the width of the autocorrelation function at the mid height. When keeping $\theta_1$ fixed and increasing $\theta_2$, we found that both populations increased their timescale *Figure 1a(i-v)*, and the ratio between the timescales of the two populations, $\tau_2/\tau_1$ did not appreciably change over a large range of time constant ratios $\theta_2/\theta_1$, *Figure 1b*. Hence, we conclude that heterogeneity in single-cell time constants does not lead to large separation of timescales in networks with strong recurrent dynamics.

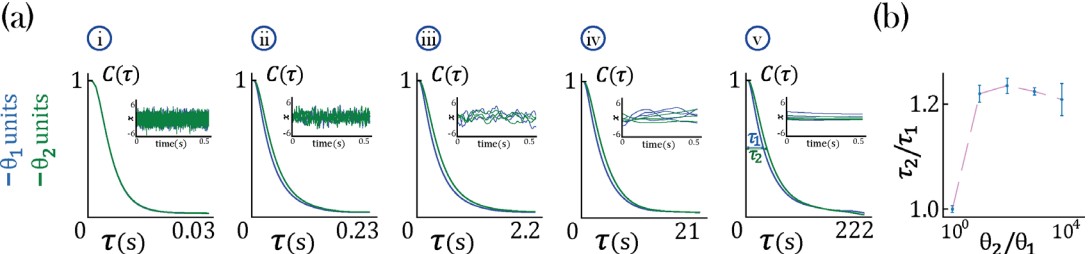

**Appendix 3—figure 1.** Timescale analysis for an RNN with two-time constants $\theta_i$, *Equation 31*, governing equal populations of neurons ($N_1 = N_2 = 1000$) and gain $g = 2.5$. (**a**) Average autocorrelation function for each population. The insert shows the dynamics of individual neurons from each population: blue for neurons with timeconstant $\theta_1$ and green for neurons with timeconstant $\theta_2$. In the networks considered here, $\theta_1 = 0.1$ ms is kept constant while: $\theta_2 = 0.1$ ms (**i**), $\theta_2 = 1.0$ ms (**ii**), $\theta_2 = 10.0$ ms (**iii**), $\theta_2 = 100.0$ ms (**iv**), $\theta_2 = 1000.0$ ms (**v**). (**b**) Population timescale ratio $\tau_2/\tau_1$ for fixed timeconstant $\theta_1 = 0.1$ ms and varying $\theta_2$.

