## [Editor Report]

This fundamental work uses computational network models to suggest a possible origin of the wide range of time scales observed in cortical activity. This claim is supported by convincing evidence based on comparisons between mathematical theory, simulations of spiking network models, and analysis of recordings from the orbitofrontal cortex. This manuscript will be of interest to the broad community of systems and computational neuroscience.

---

## [Decision Letter]

**Decision letter after peer review:**

Thank you for submitting your article "A reservoir of timescales emerges in recurrent circuits with heterogeneous neural assemblies" for consideration by *eLife*. Your article has been reviewed by 2 peer reviewers, and the evaluation has been overseen by a Reviewing Editor and Michael Frank as the Senior Editor.

---

## [Author Response]

Essential revisions:– Clarify to which extent the rate model and the spiking model rely on identical mechanisms. Tone down the claims if the correspondence is more indirect then initially suggested.

We overhauled our manuscript to present a critical assessment of both the similarities and the differences between the rate and spiking model, and performed series of theoretical investigations supported by new simulations, aimed at comparing the neural mechanisms of timescale heterogeneity between the two models. First, we introduced a mean-field-theory description of the spiking network, yielding a reduced network whose degrees of freedom are the neural assemblies themselves. In this reduced network, we identified the Schur eigenvectors of the synaptic coupling matrix as the approximate network dynamical modes, which represent strongly coupled pairs of excitatory and inhibitory assemblies. We showed that each mode is associated with a large positive eigenvalue in the synaptic matrix, and that the distribution of such gapped eigenvalues lies at the origin of the heterogeneous distribution of timescales. A similar analysis performed in the rate model suggested that the best correspondence between spiking and rate models is at the level of these Schur eigenvectors. This new analysis shows that the heterogeneity of timescales in the two model arises from a similar dynamical mechanism, namely, a heterogeneity in the eigenvalue distribution, which, in the spiking network, can be traced back to the self-couplings of neural assemblies via the mean-field reduced picture.

– Clarify the importance of coloured noise in the mechanism by adding comparisons with the white-noise case. When does the coloured noise theory in Figure 4c break down?

In the revised manuscript and in our detailed response to the Reviewers below, we elaborate on the comparison between colored noise and white noise in the probe approximation of the rate network for large self-couplings, including the analysis of several alternative models based on the white noise case, as suggested by the Reviewers. Moreover, we clarify the extent to which our Universal Colored Noise Approximation and its limit of validity. Regarding specifically the white noise-based theory, we argue that it works well for a unit with self-coupling *s*^2^
*> g*, where *g* is the network gain. The reason is that the network dynamics is responsible for generating the mean field "noise", *g*, which acts to reduce the potential barrier between the bi-stable states (and generates a concave part of the potential between its two convex spaces), while the self-coupling acts to increase the barrier and the convexity of the two bistable potential states. Since the self-coupling *s* impacts the dynamical behavior by its squared value (as manifest in the log timescale dependency and the mean-field solutions), while the network impact *g* remains linear, we estimate a good approximation at *s*^2^
*> g*.

– Show how the distribution of time scales in the spiking network depends on total network size and in-degree

In the revised manuscript, we overhauled the presentation of the spiking network model by showing how network parameters scale with the network size. We show that, in network of increasingly larger sizes, the heterogeneous distribution of timescales is maintained and its agreement with the theory improves. We further highlight how the Schur decomposition into dynamical modes described above scales with network size. Because our spiking network synaptic coupling matrix was based on Erdos-Renyi connectivity, neurons within the same neural assemblies have different in-degrees, which we chose to scale with a small linear dependence on network size.

Reviewer #2 (Recommendations for the authors):Relation between time scales and single-unit propertiesThe text in lines 223ff and Figure 4 argue that the large separation in time scales is due to an interplay between the bistability of single units due to strong self-coupling and the colored noise input from the recurrent network. The authors show that a large time scale does not emerge if the recurrent network input is replaced by white noise input (Figure 4c). How does this result go together with the following arguments: Assume low white noise input such that one can effectively linearize the dynamics. Then the self-coupling s effectively changes the time constant of the neuron (tau = 1/(1-s)). A large time scale could then be achieved by sending s -> 1.So would for example a non-chaotic regime (small g), low external white noise input and fine-tuned self-couplings be an alternative explanation for the large time scales?Appendix 2 seems to also touch upon this issue. However, there the network is in a chaotic regime such that differences in single-neuron time scales seem to get overwritten by the large colored recurrent network input. Adding a bit of white noise input to eq 28 and lowering g to operate in the regular regime would in my opinion create a large diversity in taus for different thetas.Can the authors please elaborate in much more detail on this issue and on possible alternative mechanisms to create a reservoir of time scales?

We thanks the Reviewer for suggesting potential alternative mechanisms, whose viability we will address below.

In the section "Separation of timescales in the bistable chaotic regime," our goal is to find an analytical expression for the relation between the timescales and the self-coupled units (the basic structural building blocks of the network) representing neural assemblies. We also aim to explore the reason behind this relation; i.e., we aim to understand how the presence of neural assemblies generates long timescales. As part of this aim, we replace the network input, as mimicked by the colored mean field, with white noise. By comparing the resulting dynamics, we can deduce the network’s contribution, as an input generator, in creating long time scales. For the comparison to hold, we must replace the network’s mean-field mimicking input with white noise in otherwise completely equivalent dynamical scenarios. This exercise clarifies that the long timescale emergence require a combination of both the single unit property (large self-couplings) together with the self-consistent colored noise arising from the network dynamics (because the white noise fails to generate the long timescales even in the presence of large self-couplings).

We agree with the reviewer that once other possibilities are considered, including weak network couplings and fine-tuning options, for example, long timescales in neural networks might be generated. Indeed, the Reviewer suggests interesting models, all of which use the help of external white noise to drive long timescales.

First of all, we note that one crucial ingredient missing in the alternative models proposed by the Reviewer, but which is one of the main motivation for our investigation, is that they are not self-consistent, namely, they need to postulate the presence of external noise drive with specific properties in order to sustain activity, hence, they still require an explanation for where how this drive is generated. Nevertheless, we explore here each of the ideas suggested by the reviewer, their advantage, disadvantage, and plausibility (all references to figures in the answer below refer to figures within this response document; in all the simulations used to generate the figures, *t*’s unit is 1ms, as is in the main text).

Units that are driven by low white noise and have self-coupling values approaching 1*.* The low noise drives weak activity (in magnitude; see Figure 2a, including for the dynamical equation and parameters). An effective linearized dynamics can be considered (due to the weak activity) with an effective time constant *tau* = *_/_*^1^1*−s*. A large timescale is achieved by sending *s →* 1; see 2b. For the phenomenon to occur (and the linearization to hold), the fluctuations magnitude has to be very weak (about 1% of the fixed-point value, Figure 2a), and the self-coupling must be tuned within 5% of its value for a single order of magnitude increase in timescale or 0.5% for two orders of magnitude, Figure 2c inset. Thus, in order to achieve a heterogeneity of timescales over orders of magnitudes requires extreme hierarchical fine tuning of the self-couplings. In contrast, our model achieves a large heterogeneity due to the exponential relationship between transition times and the square of the self- coupling. A Network that is operating in the non-chaotic regime, g < 1, with low external white noise and fine-tuned self-coupled units. There are a few options here for the choice of the network gain *g* in relation to the noise level. If *g <<* 1, the network drive to each unit is negligible, and the units are practically decoupled. This is effectively the scenario discussed in the previous example. If *g* is less but close to 1, yet in the non-chaotic regime, the noise fails to create long timescale effects since it is overridden by the network, whose impact by itself dies quickly. Hence, the interesting scenario is when *g* and the noise are roughly the same scale. In this case the timescales of the activity of self-coupled units with *s →* 1 are somewhat lengthened, and that propagates via the network to other units, Figure 3a. Interestingly, the timescales are strongly dependent on the realization, to the point of almost no correlation between each unit’s timescale in different network couplings realization. As a result, this is an intriguing way to generate, from white noise, via a network with self-coupled units, a correlated noise signal with randomly spread timescales.

A Network that is operating in the non-chaotic regime*, g <* 1, with low external white noise and different membrane time constants. To test this model, we split the network into two groups, each with a different membrane time constant (similar to Appendix 3, except the network here operates in the non-chaotic regime). We vary the membrane time constant of the second group. A lengthening of the timescale of the neurons in the second group is observed, Figure 3b. The lengthening (in log scale) was sublinear and converging, meaning a significant increase in the membrane time constant was needed to change the magnitude of the timescales. Moreover, the first group of neurons (with a fixed membrane constant) followed the changes in timescales, so the span of the timescales in the system remains bounded.

Mapping of clustered spiking network to network of bistable rate units- The abstract is a bit misleading as it suggests that the authors build an analytic theory for time scales in relation to assembly size. Yet, the mapping between clustered spiking networks and bistable rate units is only qualitative, and the theory only applies to the rate network. There is no quantitative theoretical prediction for the spiking network. For the purpose of the current study, this rationale is fully sufficient. The authors should just make this point more clear.

We agree the Reviewer that the analytical theory holds for rate networks, and the results of the spiking models are based on simulations only. We clarified this point explicitly in the revised abstract.

Fine-tuning needed?Can the authors elaborate more on how much fine-tuning is needed in the spiking network to obtain the large repertoire of time scales? The time scales in Figure 5b are all below 100s while the experimental data also show larger time scales. Is it possible to obtain even larger time scales in the spiking network model? What modification would be needed?

We apologize with the Reviewer for a typo in the previous version of the manuscript in Figure 3b caption: the unit for the timescales in the data and in the model is milliseconds not seconds (see also the corresponding plot in Figure 2c of (? )). We corrected the typo in the revised version. The range of timescales in the spiking network are thus well above the range observed in the data, and it would be interesting to design ad hoc experiments with chronic recordings to estimate the actual maximum range of timescales in empirical data, before circadian rhythms of secular effects come into play. Regarding the amount of fine tuning required in the model, in the revised manuscript we performed an extensive set of stimulations to investigate the scaling properties of the timescale distribution as one increases the size of the network. We found that the timescale distribution maintains a similar range when network sizes are scaled up from *N* = 2000 up to *N* = 10*,*000. In the revised manuscript, we overhauled the Methods section (e) and the Table to present the spiking network scaling parameterization. Moreover, we replaced the network in the main Figure 5 with the largest size *N* = 10*,*000, and introduced a new Appendix 2 to showcase the scaling properties of the spiking network and its Schur decomposition. Regarding potential fine tuning in other parameters, we believe that an extensive exploration of the full high-dimensional parameter space of the model is beyond the scope of the present paper, whose goal was to present an example of a spiking model that realizes the theoretical mechanism proposed in the rate model.